# Developmental function and state transitions of a gene expression oscillator in *Caenorhabditis elegans*

Milou WM Meeuse[1,2,†] (iD), Yannick P Hauser[1,2,†] (iD), Lucas J Morales Moya[1], Gert-Jan Hendriks[1,2] (iD), Jan Eglinger[1] (iD), Guy Bogaarts[3] (iD), Charisios Tsiairis[1] (iD) & Helge Großhans[1,2,*] (iD)

## Abstract

**Gene expression oscillators can structure biological events temporally and spatially. Different biological functions benefit from distinct oscillator properties. Thus, finite developmental processes rely on oscillators that start and stop at specific times, a poorly understood behavior. Here, we have characterized a massive gene expression oscillator comprising > 3,700 genes in *Caenorhabditis elegans* larvae. We report that oscillations initiate in embryos, arrest transiently after hatching and in response to perturbation, and cease in adults. Experimental observation of the transitions between oscillatory and non-oscillatory states at high temporal resolution reveals an oscillator operating near a Saddle Node on Invariant Cycle (SNIC) bifurcation. These findings constrain the architecture and mathematical models that can represent this oscillator. They also reveal that oscillator arrests occur reproducibly in a specific phase. Since we find oscillations to be coupled to developmental processes, including molting, this characteristic of SNIC bifurcations endows the oscillator with the potential to halt larval development at defined intervals, and thereby execute a developmental checkpoint function.**

**Keywords** bifurcation; checkpoint; development; oscillator; SNIC
**Subject Category** Development & Differentiation
**Mol Syst Biol. (2020) 16: e9498**

## Introduction

Gene expression oscillations occur in many biological systems as exemplified by circadian rhythms in metabolism and behavior (Panda *et al*, 2002), vertebrate somitogenesis (Oates *et al*, 2012), plant lateral root branching (Moreno-Risueno *et al*, 2010), and *Caenorhabditis elegans* larval development (Hendriks *et al*, 2014). They are well-suited for timekeeping, acting as molecular clocks that can provide a temporal, and thereby also spatial, structure for biological events (Uriu, 2016). This structure may represent external time, as illustrated by circadian clocks, or provide temporal organization of internal processes without direct reference to external time, as illustrated by somitogenesis clocks (Rensing *et al*, 2001).

Depending on these distinct functions, oscillators require different properties. Thus, robust representation of external time requires a stable period; i.e., the oscillator has to be compensated for variations in temperature and other environmental factors. It also benefits from a phase-resetting mechanism to permit realignments, if needed, to external time. Intuitively, either feature seems unlikely to benefit developmental oscillators. By contrast, because developmental processes are finite; e.g., an organism has a characteristic number of somites, developmental oscillators need a start and an end. How such changes in oscillator activity occur *in vivo*, and which oscillator features enable them, is largely unknown (Riedel-Kruse *et al*, 2007; Shih *et al*, 2015).

Here, we characterize the recently discovered "*C. elegans* oscillator" (Kim *et al*, 2013; Hendriks *et al*, 2014) at high temporal resolution and across the entire period of *C. elegans* development, from embryo to adult. The system is marked by a massive scale where ~3,700 genes exhibit transcript level oscillations that are detectable, with large, stable amplitudes and widely dispersed expression peak times (i.e., peak phases), in lysates of whole animals. For the purpose of this study, and because insufficient information exists on the identities of core oscillator versus output genes, we define the entire system of oscillating genes as "the oscillator". We demonstrate that the oscillations are coupled to molting, i.e., the cyclical process of new cuticle synthesis and old cuticle shedding that occurs at the end of each larval stage. We observe and characterize onset and offset of oscillations both during continuous development and upon perturbation, and find that transitions occur with a sudden change in amplitude. They also occur in a characteristic oscillator phase and thus at specific, recurring intervals. The transitions are a manifestation of a bifurcation, i.e., qualitative change in behavior, of the underlying oscillator system. Hence, our observations constrain possible oscillator

1 Friedrich Miescher Institute for Biomedical Research (FMI), Basel, Switzerland
2 University of Basel, Basel, Switzerland
3 University Hospital, Basel, Switzerland
*Corresponding author. Tel: +41 61 697 6580; E-mail: helge.grosshans@fmi.ch
†These authors contributed equally to this work
Published research reagents from the FMI are shared with the academic community under a Material Transfer Agreement (MTA) having terms and conditions corresponding to those of the UBMTA (Uniform Biological Material Transfer Agreement).

architectures, excluding a simple negative-loop design, and parametrization of mathematical models.

Functionally, because of the phase-locking of the oscillator and molting, arrests always occur at the same time during larval stages, around molt exit. This time coincides with the previously reported recurring window of activity of a checkpoint that can halt larval development in response to nutritionally poor conditions. Hence, our results indicate that the *C. elegans* oscillator functions as a developmental clock whose architecture supports a developmental checkpoint function.

# Results

### Thousands of genes with oscillatory expression during the four larval stages

Although previous reports agreed on the wide-spread occurrence of oscillatory gene expression in *C. elegans* larvae (Kim *et al*, 2013; Grün *et al*, 2014; Hendriks *et al*, 2014), the published datasets were either insufficiently temporally resolved or too short to characterize oscillations across *C. elegans* larval development. Hence, to understand the extent and features of these oscillations better, including their continuity throughout development, we performed two extended time course experiments to cover the entire period of post-embryonic development plus early adulthood at hourly resolution. We extracted total RNA from populations of animals synchronized by hatching in the absence of food. The first time course (designated TC1) covered the first 15 h of development on food at 25°C, and the second time course (TC2) covered the span of 5 h through 48 h after plating at 25°C. [Fig EV1A provides a summary of all sequencing time courses analyzed in this study.] The extensive overlap facilitated fusion of these two time courses into one long time course (TC3) (Fig EV1B), and a pairwise-correlation plot of gene expression over time showed periodic similarity (Fig 1A, light-gray off-diagonals).

The larger dataset enabled us to improve on the previous identification of genes with oscillatory expression (Hendriks *et al*, 2014). Using cosine wave fitting, and an amplitude cut-off of $\log_2$(amplitude) $\geq 0.5$ and $P \leq 0.01$, we classified 3,739 genes (24% of total expressed genes) as "oscillating" (i.e., rhythmically expressed) from

TC2 (Figs 1B and EV1C and Dataset EV1; Materials and Methods). We confirmed this classification using MetaCycle (Wu *et al*, 2016, 2019), an algorithm that is widely used to study rhythmic circadian gene expression. At an FDR < 0.05 and an amplitude cut-off of $\log_2$(amplitude) $\geq 0.5$, MetaCycle identified a comparable number, and highly overlapping set, of oscillating genes with similar amplitudes (Appendix Fig S1A and B). It also confirmed a predominant 7-h period for these genes (Appendix Fig S1C). We conclude that cosine fitting works robustly to identify oscillating genes in our data.

Relative to the previous result of 2,718 oscillating genes (18.9% of total expressed genes) in mRNA expression data of L3 and L4 animals (Hendriks *et al*, 2014), this adds 1,240 new genes and excludes 219 of the previously annotated oscillating genes. We consider this latter group to be most likely false positives from the earlier analysis, resulting from the fact that some genes behave substantially different during L4 compared to the preceding stages as shown below.

Visual inspection of a gene expression heatmap of the fused time course (TC3; Fig 1C) revealed four cycles of gene expression for the oscillating genes, presumably reflecting progression through the four larval stages. Oscillations were absent during the first few hours of larval development as well as in adulthood, from ~37 h on, and both their onset and offset appeared to occur abruptly. We will analyze these and additional features of the system and their implications in more detail in the following sections.

### Oscillating genes are expressed in several tissues with dispersed peak phases

An examination of the calculated peak phases confirmed the visual impression that individual transcripts peaked at a wide variety of time points, irrespective of expression amplitude (Fig 1D). In circadian rhythms, peak phase distributions are typically clustered into three or fewer groups when examined in a specific tissue (Koike *et al*, 2012; Korenčič *et al*, 2014). However, the identity of oscillating genes differs across cell types and tissues, and for those genes that oscillate in multiple tissues, phases can differ among tissues (Zhang *et al*, 2014). Hence, we wondered whether the broad peak phase distribution was a consequence of our analysis of RNA from whole animals, whereas individual tissues might exhibit a more defined phase distribution.

**Figure 1. A massive oscillator with dispersed peak phases in several tissues.**

A   Pairwise correlation plot of $\log_2$-transformed expression patterns of all genes (*n* = 19,934) obtained from a synchronized population of L1 stage larvae sampled and sequenced from *t* = 1 h until *t* = 48 h (TC3; a fusion of the two time courses TC1 and TC2 after 13 h; Fig EV1A and B). An asterisk indicates an outlier, time point *t* = 40 h.

B   Scatter plot identifying genes with oscillatory expression (henceforth termed oscillating genes, *blue*) based on amplitude and 99% confidence interval (99%-CI) of a cosine fitting of their expression quantified on TC2 (Materials and Methods). A lower CI-boundary $\geq 0$, i.e., $P \leq 0.01$, and a $\log_2$(amplitude) $\geq 0.5$, which corresponds to a 2-fold change from peak to trough, were used as cut-offs. Genes below either cut-off were included in the "not oscillating" group (*black*). Figure EV1C shows gene distributions in a density scatter plot.

C   Gene expression heatmap of oscillating genes as classified in Figs 1B and EV1C. Oscillating genes were sorted by peak phase, and mean expression per gene from *t* = 7 h to *t* = 36 h (when oscillations occur) was subtracted. *n* = 3,680 as not all genes from the long time course (TC2) were detected in the early time course (TC1). Gray horizontal bars indicate the individual oscillation cycles, C1 through C4 which start at TP6, TP14, TP20, and TP27, respectively, as later determined in Appendix Fig S7.

D   Radar chart plotting amplitude (radial axis, in $\log_2$) over peak phase (circular axis, in degrees) as determined by cosine fitting in Fig 1B.

E   Enrichment (*red*) or depletion (*blue*) of tissues detected among oscillating genes expressed tissue-specifically relative to all tissue-specific genes using annotations derived from Cao *et al* (2017). Significance was tested using one-sided binomial tests which resulted in *P*-values < 0.001 for all tissues.

F   Density plot of the observed peak phases of tissue-specifically expressed oscillating genes for all enriched tissues.

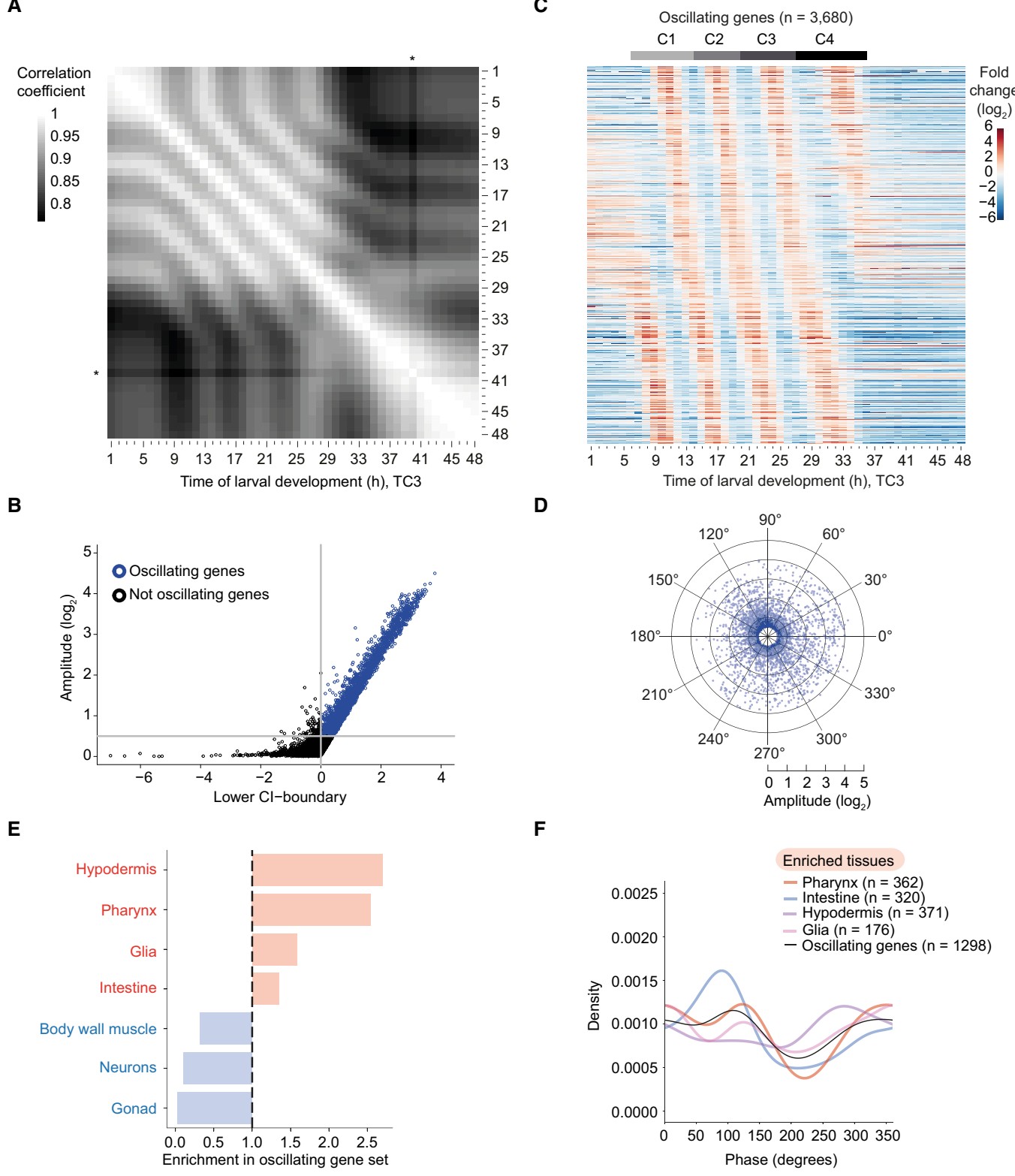

**Figure 1**

To understand in which tissue oscillations occur, we utilized a previous annotation of tissue-specifically expressed genes (Cao *et al*, 2017). 1,298, and thus a substantial minority (~35%) of oscillating genes, fell in this category for seven different tissues. They were strongly (~2.5-fold) enriched in the hypodermis (epidermis) and pharynx, and more modestly (≤ 1.5-fold) in glia and

intestine (Fig 1E). By contrast, oscillating genes were greatly depleted from body wall muscle, neurons, and gonad. Hence, oscillatory gene expression occurs indeed in multiple tissues. However, although peak phase distributions deviated for each tissue to some degree from that seen for all oscillating genes, they were still widely distributed for each individual tissue (Fig 1F).

We conclude that a wide dispersion of peak phases appears to be an inherent oscillator feature rather than the result of a convoluted output of multiple, tissue-specific oscillators with distinct phase preferences.

### Oscillations initiate with a time lag in L1

The observation that oscillations were undetectable during the first few hours of larval development and started only after > 5 h into L1 (Fig 1A and C) surprised us. Hence, we performed a separate experiment that covered the first 24 h of larval development (TC4). This confirmed our initial finding of a lack of oscillations during the first few hours of larval development (Fig 2A and B).

To understand how oscillations initiate after the initial quiescence, we looked at individual genes and observed that the start of detectable oscillations differed for individual genes (Fig 2A and C). Nonetheless, the occurrence of first peaks was globally well correlated with the peak phases calculated from data in Fig 1 (Fig 2D); i.e., the order in which the first peaks of gene expression occurred was the same as the order given by the peak phases calculated for the stably running oscillator. (The apparent discontinuity in the data at a first peak of ~10 h is explained by the circularity of the data, with $0° = 360°$, and the arbitrary assignment of $0°$ in the peak phase calculation.) Moreover, the transcript levels of many genes with a late-occurring (11–13 h) first peak proceeded through a trough before reaching their first peak as exemplified in Fig 2C for *F11E6.3*. We conclude that oscillations exhibit a structure of phase-locked gene expressing patterns as soon as they become detectable.

### L1 larvae undergo an extended intermolt

Although the gene expression oscillations occur in the context of larval development, functional connections have been lacking. However, genes encoding cuticular components were reported to be enriched among previously identified oscillating genes (Kim *et al*, 2013; Hendriks *et al*, 2014), and Gene Ontology (GO-) term analysis of the new extended set of oscillating genes confirms that the top 12 enriched terms all linked to cuticle formation and molting, or protease activity (Fig 3A, Dataset EV2). These findings, and the fact that molting is itself a rhythmic process, repeated at the end of each larval stage, suggest the possibility of a functional link between molting and gene expression oscillations.

If such a link were true, we would predict that the initial period of quiescence in the early L1 stage be accompanied by a lengthened stage, and, specifically, an extended intermolt duration. Indeed, using a luciferase-based assay that reveals the period of behavioral quiescence, or lethargus, that is associated with the molt (Appendix Fig S2A and B), others had previously reported an extended L1 relative to other larval stages (Olmedo *et al*, 2015). However, they reported an extension of both molt and intermolt.

As the previously used luciferase-expressing transgenic strains developed relatively slowly and with limited synchrony across animals, presumably due to their specific genetic make-up, we repeated the experiment with a newly generated strain that expressed luciferase from a single-copy integrated transgene and that developed with improved synchrony and speed (Fig EV2A and B, Appendix Fig S2E–G, Materials and Methods). Our results confirmed that L1 was greatly extended relative to the other larval stages (Fig EV2E). However, in contrast to the previous findings (Olmedo *et al*, 2015), but consistent with our hypothesis, the differences appeared largely attributable to an extended intermolt (Fig EV2D). The duration of the first molt (M1) was instead comparable to that of M2 and M3 (Fig EV2C).

Thus, an extended first intermolt coincides with the fact that no oscillator activity can be detected by RNA sequencing during the first 5 h of this larval stage. Moreover, because we performed the experiment by hatching embryos directly into food, we can conclude that the extended L1 stage is an inherent feature of *C. elegans* larval development, rather than a consequence of starvation-induced synchronization.

### Development is coupled to oscillatory gene expression

The luciferase assay revealed that also the L4 stage took significantly longer than the two preceding stages, though not as long as L1 (Fig EV2E). In this case, both the fourth intermolt and the fourth molt were extended (Fig EV2C and D). As apparent from the gene expression heatmap, and quantified below, the oscillation period during L4 was also extended. Hence, grossly similar trends appeared to occur in larval stage durations and oscillation periods, determined by the luciferase assay and RNA sequencing, respectively. We considered this as further evidence for a coupling of the two processes.

To test this hypothesis explicitly, we sought to quantify the synchrony of oscillatory gene expression and developmental progression in individual animals at the same time. To this end, we established a microchamber-based time-lapse microscopy assay by adapting a previous protocol (Turek *et al*, 2015). In this assay, animals are hatched and grown individually in small chambers where they can be tracked and imaged while moving freely, enabling their progression through molts. Using Mos1-mediated single-copy transgene integration (MosSCI) (Frøkjær-Jensen *et al*, 2012), we generated transgenic animals that expressed destabilized *gfp* from the promoter of *qua-1*, a highly expressed gene with a large mRNA level amplitude.

Consistent with the RNA sequencing data, we detected oscillations of the reporter with four expression peaks (Fig 3B). Moreover, we observed similar rates of development as in the luciferase assays when we curated the molts (Fig 3C, Appendix Table S1, Materials and Methods). Using a Hilbert transform (Pikovsky *et al*, 2001) to quantify the instantaneous, i.e., time-varying, changes in period (see Appendix), we observed that the averaged reporter oscillation period times for each cycle were in good agreement with the stage durations, for all three larval stages, L2-L4, for which oscillations period lengths could be reliably determined (Fig 3D).

Single animal imaging enabled us to ask when molts occurred relative to oscillatory gene expression, and we observed a very uniform behavior across animals (Fig 3B, red segments). To quantify this relationship, we determined the gene expression phases at molt entries and exits. We obtained highly similar values across worms within one larval stage (Fig 3E), and only a minor drift when comparing phases across larval stages. Accordingly, when we

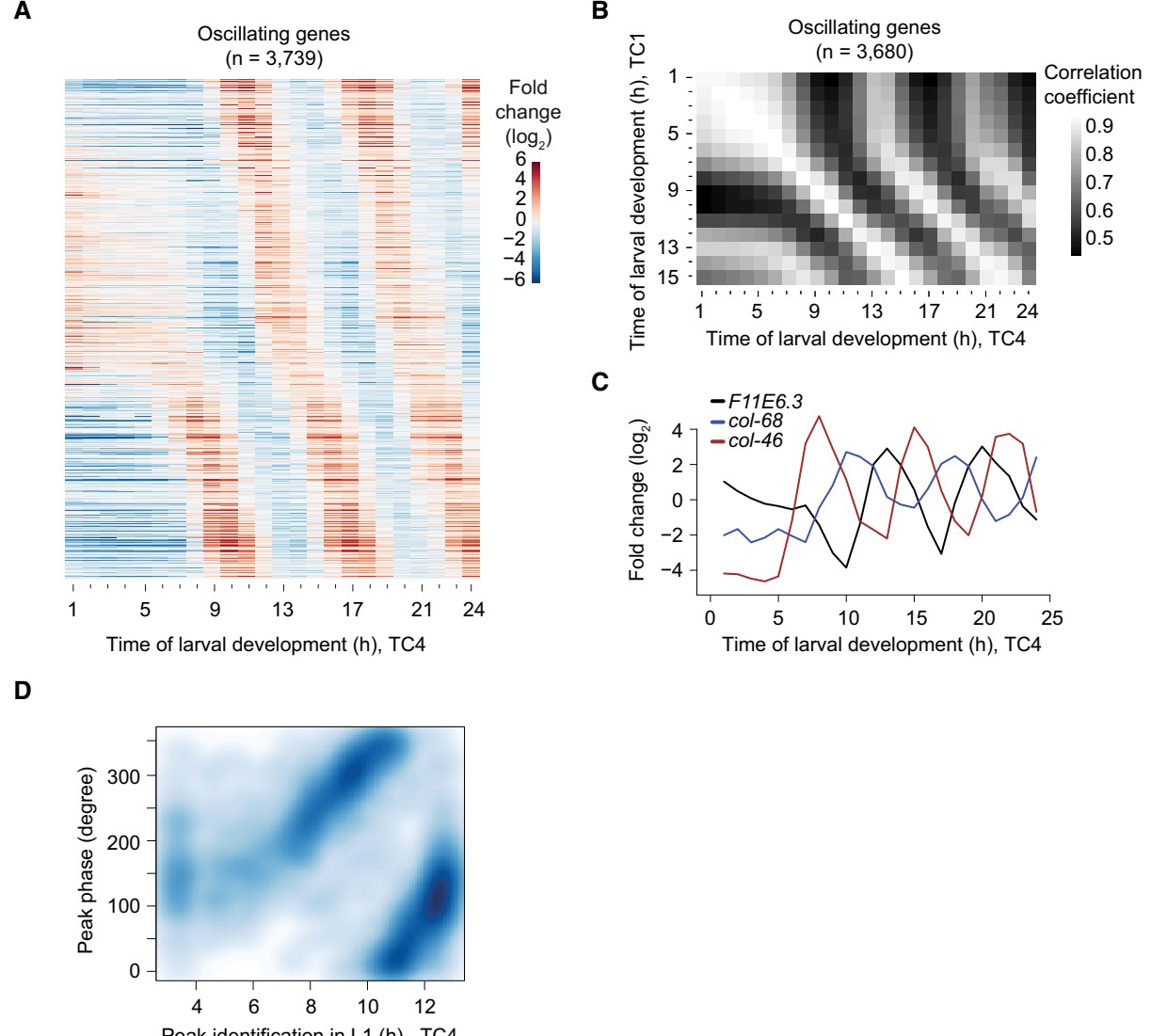

**Figure 2. Oscillations start with a time lag in L1.**

A  Gene expression heatmap of detectably expressed oscillating genes sampled from a separate early developmental time course (TC4; $t = 1$ h to $t = 24$ h). Genes were ranked according to their peak phase determined in Fig 1.

B  Pairwise correlation plot of $log_2$-transformed oscillating gene expression data obtained from both early larval development time courses, TC1 and TC4.

C  Gene expression traces of the representative genes *F11E6.3*, *col-68*, and *col-46*.

D  Scatter plot of calculated oscillating gene peak phase (as determined in Fig 1) over the time of occurrence of the first expression peak in L1 larvae, observed in TC4. Peak detection was performed using a spline analysis. As visual inspection did not reveal peaks in the heatmap during the first 3 h, and as the first cycle ends at 13 h, we performed this analysis for $t = 3$ h to $t = 13$ h to reduce noise. Generally, peak phases determined from the full developmental time course (TC3) correlate with the time of the first peak in L1.

Data information: All analyses for oscillating genes identified in Fig 1 with detectable expression ($n = 3,739$ in A, $n = 3,680$ in B).

plotted the times required to reach a specific, arbitrarily chosen oscillation phase in L2 or L3 over the time required to reach M2 or M3, we observed high levels of correlation ($r > 0.9$ for each instance; Fig 3F, Appendix Fig S3). Two additional reporter transgenes, based on the promoters of *dpy-9* and *F11E6.3*, which differ in peak expression phases from *qua-1* and one another, yielded similar results (Appendix Fig S4).

We considered two possible interpretations of the close correlation between developmental progression and oscillations: First,

both oscillations and development could be under independent, but precise temporal control. In this model, certain developmental events would merely coincide with specific phases of oscillations rather than being coupled to them. Therefore, variations in the periods of oscillation and development would add up, non-linearly, to the experimentally observed phase variations. Second, phase-locking of oscillatory gene expression and developmental events might result from the two processes being truly coupled and/or from one driving the other. In this case, the variations in

the two periods would partially explain each other, causing a reduction in the expected phase variation relative to the first scenario (Fig 3G).

To distinguish between these scenarios, we used error-propagation to calculate the expected error for two independent processes (Materials and Methods). Focusing on L2 and L3 stages to exclude

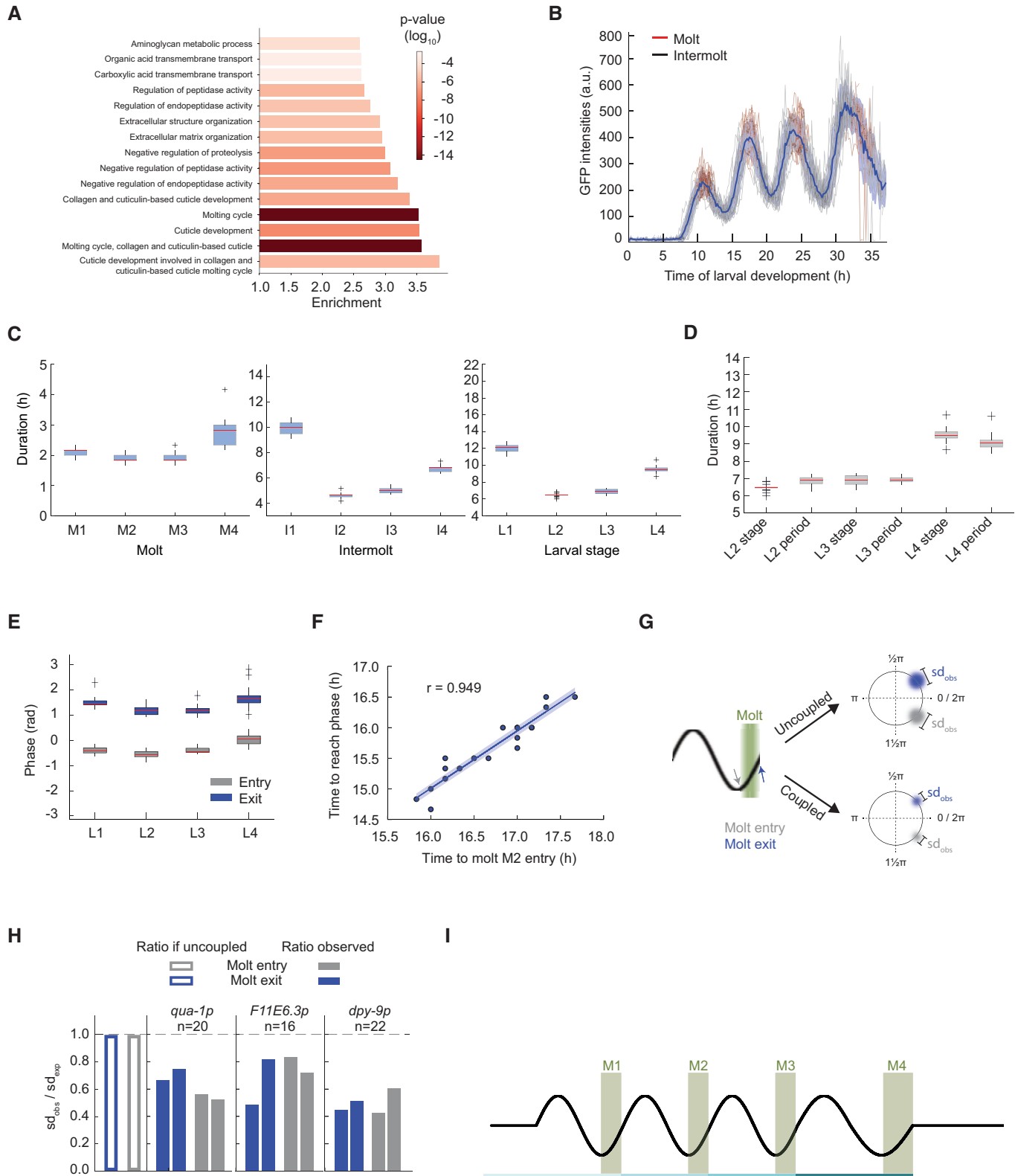

**Figure 3.**

**Figure 3. Oscillatory gene expression is coupled to molting.**

A GO-term enrichments for oscillating genes as classified in Fig 1C. *P*-values were calculated using Fisher's exact test. The top 15 enriched terms are displayed.

B GFP signal quantification for *qua-1p::gfp::pest::h2b::unc-54*$_{3'UTR}$ expressing single animals (HW2523, *n* = 20) over larval development, starting from hatch (*t* = 0 h). Individual traces are colored in *black* during the intermolt and in *red* during the molt. The mean intensity (*blue line*) and standard deviation across population (*shading*) are indicated.

C, D Boxplots of molt, intermolt, and larval stage durations (C) and of larval stage durations and period times of oscillations (D) of single animals (HW2523) developing in microchambers (*n* = 20). In (D), L1 was excluded because of the time lag before oscillations manifest after hatching.

E Boxplot of phase at molt entry (start of lethargus) and molt exit (end of lethargus) separated by larval stages for single animals (HW2523) developing in microchambers (*n* = 20)

F Scatterplot comparing developmental duration until second molt entry (M2 entry) with time to reach an arbitrarily chosen, unwrapped GFP oscillation phase obtained from data in (B). The particular phase chosen was observed close to the end of L2. The Pearson correlation is indicated with "r".

G Schematic model of expected phase variation at molt entry (*gray*) and molt exit (*blue*) depending on the coupling status between oscillations and molting. Width of colored blur is proportionate to observed standard deviation, sd$_{obs}$, with sd$_{obs,uncoupled}$ > sd$_{obs,coupled}$.

H Barplots displaying the ratio of observed standard deviation over expected standard deviation for phase calling from GFP intensity oscillations as measured in B, at either molt entry or molt exit for the indicated reporters. The empty bars indicate the expected value in the case of uncoupled processes $\left(\frac{sd_{obs}}{sd_{exp}}=1\right)$. For coupled processes, we would expect $\frac{sd_{obs}}{sd_{exp}}<1$. The $\frac{sd_{obs}}{sd_{exp}}$ values for all reporters are below 1. A dashed line indicates parity. (See Materials and Methods)

I Schematic depiction of coordination between oscillatory gene expression and development.

Data information: Boxplots in (C–E) extend from first to third quartile with a line at the median, outliers are indicated with a cross, and whiskers show 1.5*IQR.

any edge effects on period calculation by Hilbert transform, we found that the observed error was consistently well below this expected (calculated) error (Fig 3H), for all three reporter genes, for both molt entry and molt exit, and for both larval stages. Thus, our observations agree with the notion that development and oscillatory gene expression are functionally coupled (Fig 3I), and potentially causally connected.

## Quantification of amplitude and period behaviors over time reveals characteristic system properties

Consistent with the coupling between oscillations and development, the last larval stage and the period of the last oscillation cycle both appeared increased (Fig 3D) before oscillations ceased. The characteristics of such a transition from oscillatory to non-oscillatory state are determined by the behavior of the underlying system, which can be understood in the light of bifurcation theory. Bifurcation, that is, a qualitative change in system behavior, occurs in response to a change in one or more control parameters. Depending on the system's topology, characteristic changes of amplitude and period occur as the bifurcation parameter changes during bifurcation (Fig 4A) (Izhikevich, 2000; Strogatz, 2015; Salvi *et al*, 2016). Thus, transition into a quiescent state through a supercritical Hopf bifurcation involves a declining, and ultimately undetectable, amplitude and a constant period. By contrast, a Saddle Node on Invariant Cycle (SNIC) bifurcation results in a declining frequency (and thus increasing period) but a stable amplitude.

Hence, to gain a better understanding of the oscillator's bifurcation, we quantified oscillation amplitudes and periods over time. To minimize variations from differences among experiments, we did this for the contiguous 5- to 48-h time course (TC2). This enabled reliable quantification of these features for the last three oscillation cycles, C2 through C4, which begin at 14 h (C2), 20 h (C3), and 27 h (C4), respectively (Fig 1C and see below). Excluding a small set of 291 genes that exhibited unusual expression trends during the fourth larval stage, i.e., a major change in mean expression levels (Appendix Fig S5), cosine fitting on the individual cycles revealed a good agreement of amplitudes across stages, and in particular no indication of damping during the last cycle, C4 (Fig 4B).

We used a Hilbert transform to quantify the period over time with high temporal resolution, i.e., at every hour of development.

The mean oscillation period thus calculated was approximately 7 h during the first cycles but increased during the fourth cycle (Fig 4C), consistent with the single worm imaging results. This change was also apparent when we reconstructed an oscillation from the mean observed oscillation period and compared it to an oscillation with a constant period of 7 h (Fig 4D).

In summary, these analyses reveal a sudden loss of oscillation upon transition to adulthood without prior amplitude damping and an oscillator that can maintain a stable amplitude in the presence of period changes. These are features of an oscillator operating near a SNIC rather than a supercritical Hopf bifurcation (Fig 4A) (Izhikevich, 2000; Strogatz, 2015; Salvi *et al*, 2016).

## Arrest of the oscillator in a specific phase upon transition to adulthood

The observation of a period extension during the L4 stage cannot be explained by a supercritical Hopf bifurcation (Fig 4A). However, as little or no period change is observed during oscillation onset in the L1 stage relative to later cycles, we lack such strong evidence against a supercritical Hopf bifurcation. This may be caused by different dynamics of the parameter change: In a SNIC bifurcation, a period change is only observed if the bifurcation parameter changes slowly, but not if it changes immediately (Fig EV3A–C). Similarly, for a supercritical Hopf bifurcation, an immediate change in bifurcation parameter would potentially allow rapid adoption of a constant amplitude (Fig EV3D).

Although experimentally observed transitions may thus not exhibit characteristic period or amplitude changes, SNIC and supercritical Hopf bifurcations differ additionally in a feature that is independent of the rate of bifurcation parameter change, namely in the stable state, or fixed point, that the systems adopt when oscillations do not occur. In a supercritical Hopf bifurcation, the system spirals from a limit cycle onto a fixed point, whereas in a SNIC bifurcation, the fixed point emerges on the limit cycle (Fig 5A) (Saggio *et al*, 2017). In other words, a quiescent oscillator near a SNIC bifurcation adopts a state similar to that of a specific phase of the oscillator; the oscillator has become "arrested". By contrast, following a supercritical Hopf bifurcation, the oscillator adopts a stable state that is distinct from any phase of the oscillator. Hence, if the *C. elegans* oscillator entered an arrested state through a SNIC bifurcation, the

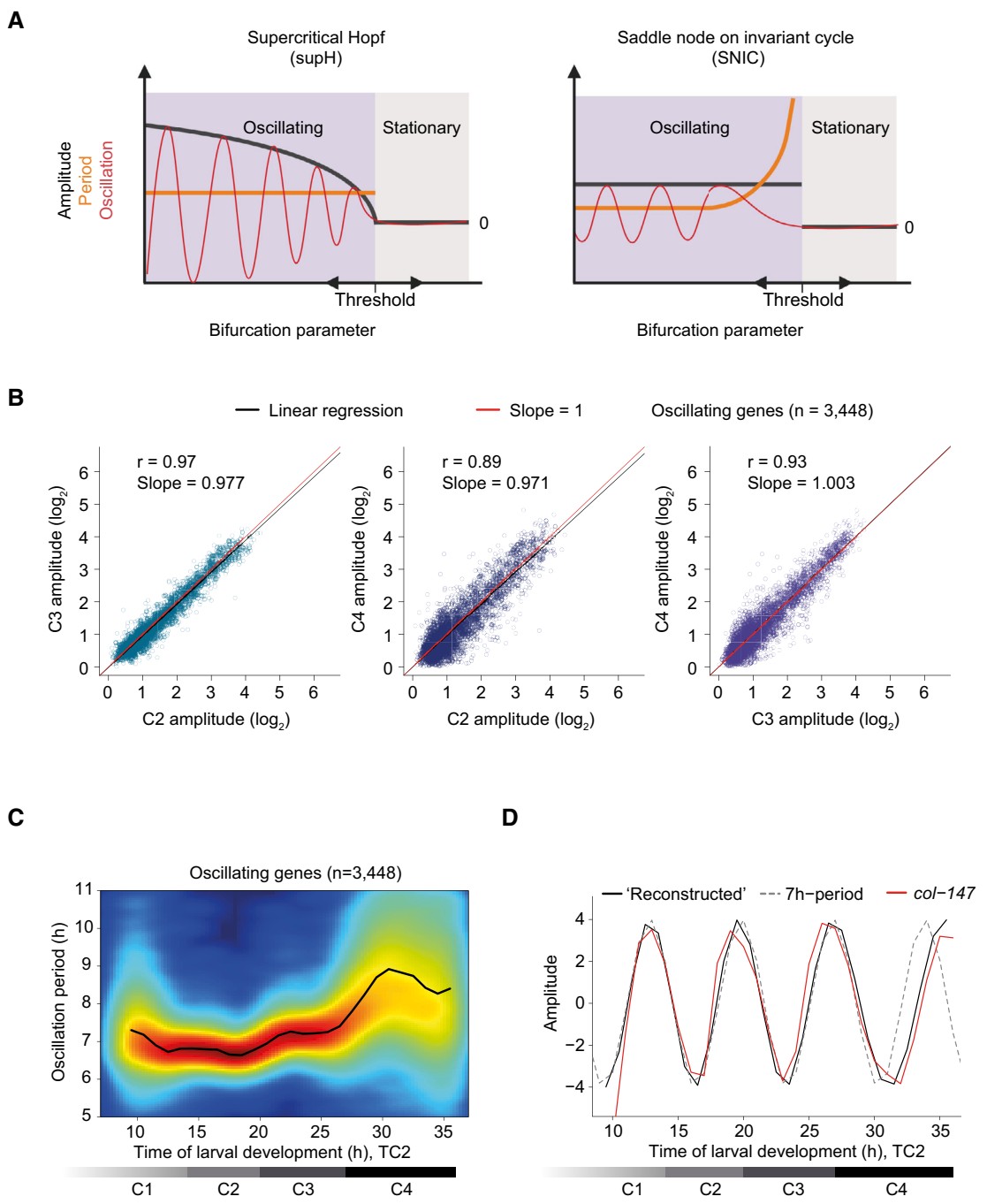

**Figure 4. Change in period without noticeable change in amplitude.**

A   Schematic depiction of amplitude and period behaviors in response to a control parameter change for an oscillatory system transitioning between a quiescent (stationary) and an oscillatory state through the indicated bifurcations (created with BioRender.com). Note that transitions can occur in either direction.

B   Amplitudes derived from cosine fitting to the individual oscillations of C2, C3, and C4 (TC2) plotted against each other. Note that the last time point of one cycle coincides with the first time point of the following one since 0° = 360°. Pearson correlation coefficient r, slope of the linear regression (*black*), and the diagonals (slope = 1; *red*) are indicated. 291 genes were excluded from oscillating genes due to altered mean expression in L4, see Appendix Fig S5, i.e., *n* = 3,448.

C   Density plot showing oscillation period at every time point for each of the oscillating genes (*n* = 3,448) as quantified by Hilbert transform. Smoothed color density, obtained through a kernel density estimate, provides a qualitative view of gene number and increases from blue to red. Mean oscillation period over all oscillating genes is shown by the black line.

D   Expression changes for an oscillation with a constant 7-h period (*dotted line*), and an oscillation reconstructed from the mean oscillation period in (C) (*black line*), both amplitudes set to four. The expression of a representative gene, *col-147* (mean normalized), is shown (*red line*).

Data information: Horizontal gray bars indicate oscillation cycles C1 through C4 which start at TP6, TP14, TP20, and TP27 as determined in Appendix Fig S7.

overall expression profile of the oscillating genes in the adult stage should resemble that seen at some other time point during larval development.

To test this prediction, we analyzed the correlation of oscillating gene expression for adult time points (TP ≥ 37 h) to all other time points of the fused time course (TC3). (In the following, we will use "TPx" to refer to any time point "x", in hours, after hatching. Technically, this is defined in our experiment as the time after plating synchronized, first larval stage animals on food.) For this analysis

(illustrated in Fig EV4), we used the pairwise correlation matrix resulting from the oscillating gene set without the previously excluded genes that changed in expression in the L4 stage (Fig 5B). This provided two insights. First, correlation coefficients among adult time points all exceeded 0.8 with little change over time, confirming the high similarity of samples TP37 – 48 to one another and thus an absence of detectable oscillations. Second, in addition to one another, TP37 – 48 are particularly highly correlated with a specific time—and thus phase—of the oscillatory regime, namely

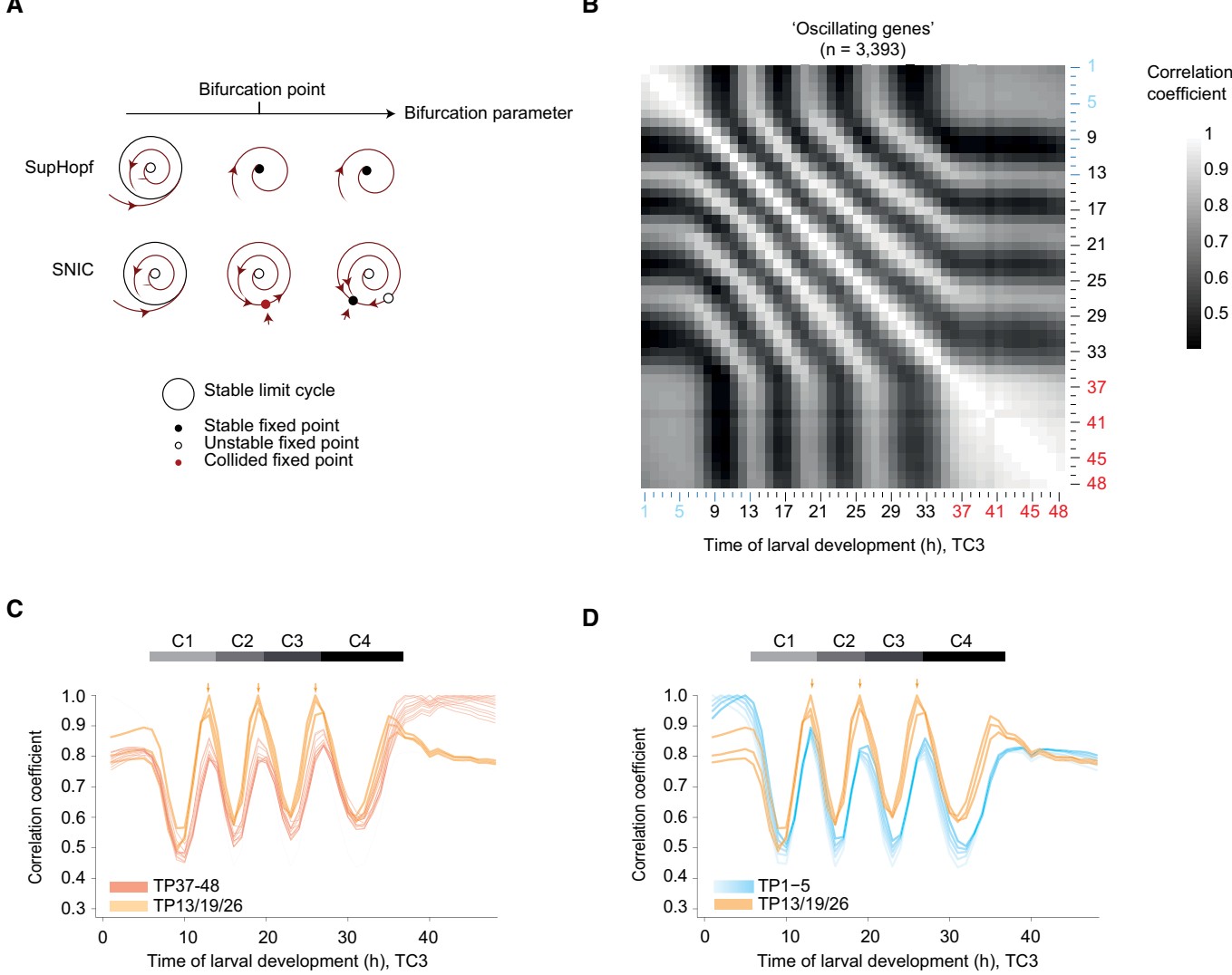

**Figure 5. The oscillator is phase arrested in early L1 and adults.**

A   Phase plane diagrams depicting supercritical Hopf (SupHopf) and SNIC bifurcations, respectively, showing the change in qualitative behavior as the bifurcation parameter value changes (arrow). The bifurcation point, i.e., the parameter value at which the bifurcation occurs, is indicated.

B   Pairwise correlation plot of log₂-transformed oscillating gene expression data obtained from TC3, i.e., the fusion of TC1 (*blue labels*) and TC2 (*black/red*). Genes which deviated in mean expression in L4 were excluded (Appendix Fig S5), resulting in *n* = 3,393 genes.

C   Lines of correlation for TP37–48 (*red*) to all time points in the fused larval time course. Arrows indicate local correlation maxima at TP13, 19, and 26. The correlation traces for TP13/19/26 are shown in orange. Figure EV4 illustrates how correlation lines were generated.

D   Lines of correlation for TP1–5 (*blue*) and TP13/19/26 (*orange*) to all time points in the fused larval time course. Arrows indicate local correlation maxima at TP13, 19, and 26.

Data information: All correlations were determined by Pearson correlation. Correlation lines plotted in (C, D) correspond to Fig 5B. Horizontal gray bars indicate oscillation cycles C1 through C4 which start at TP6, TP14, TP20, and TP27 as determined in Appendix Fig S7.

TP13 and the "repetitive" TP19 and TP26 (Fig 5C, arrows). In other words, expression levels of oscillating genes in the adult resemble a specific larval oscillator phase, providing further support for a SNIC bifurcation.

### Phase-specific arrest of the oscillator after hatching

We noticed that the gene expression states of TP37 – 48 also correlated well to each of TP1 – 5; i.e., the early L1 larval stage before oscillations is detected (Fig 5C). To examine this further, we performed the same correlation analysis as described above, but now for TP1 – 5. Mirroring the adult situation, correlation coefficients among all these five time points were high and exhibited little change over time, and TP1 – 5 exhibited particularly high levels of correlation with TP13, TP19, and TP26 (Fig 5D). These are the same larval time points to which the adult time points exhibit maximum similarity. We confirmed these two key observations when fusing the independent time course TC4 to TC2 (generating TC5; Appendix Fig S6).

We conclude that also during the first 5 h after plating, oscillating genes adopt a stable expression profile that resembles a specific

phase of the oscillator. We note that a Hopf bifurcation cannot yield ready phase control, i.e., initiation of the oscillator from a specific phase, in particular in the presence of noise (Fig EV3G). Hence, these findings provide further support for the notion that both oscillation onset and offset through a SNIC bifurcation. Indeed, this explains our observation (Fig 2) that in L1 stage larvae, oscillations exhibit a structure of phase-locked gene expressing patterns as soon as they become detectable: The oscillator initiates from an arrested phase.

### Initiation of oscillation soon after gastrulation

We wondered how the oscillator entered the arrested state observed in early larvae, i.e., what dynamics the class of larval oscillating genes exhibited in embryos. Hence, we examined single embryo gene expression data from a published time series (Hashimshony *et al*, 2015). When plotting the embryonic expression patterns of oscillating genes sorted by their peak phase defined in larvae, we observed a dynamic expression pattern with a striking phase signature (Fig 6A). To investigate this further, we performed a correlation analysis between embryonic and larval time points (TC3) for the

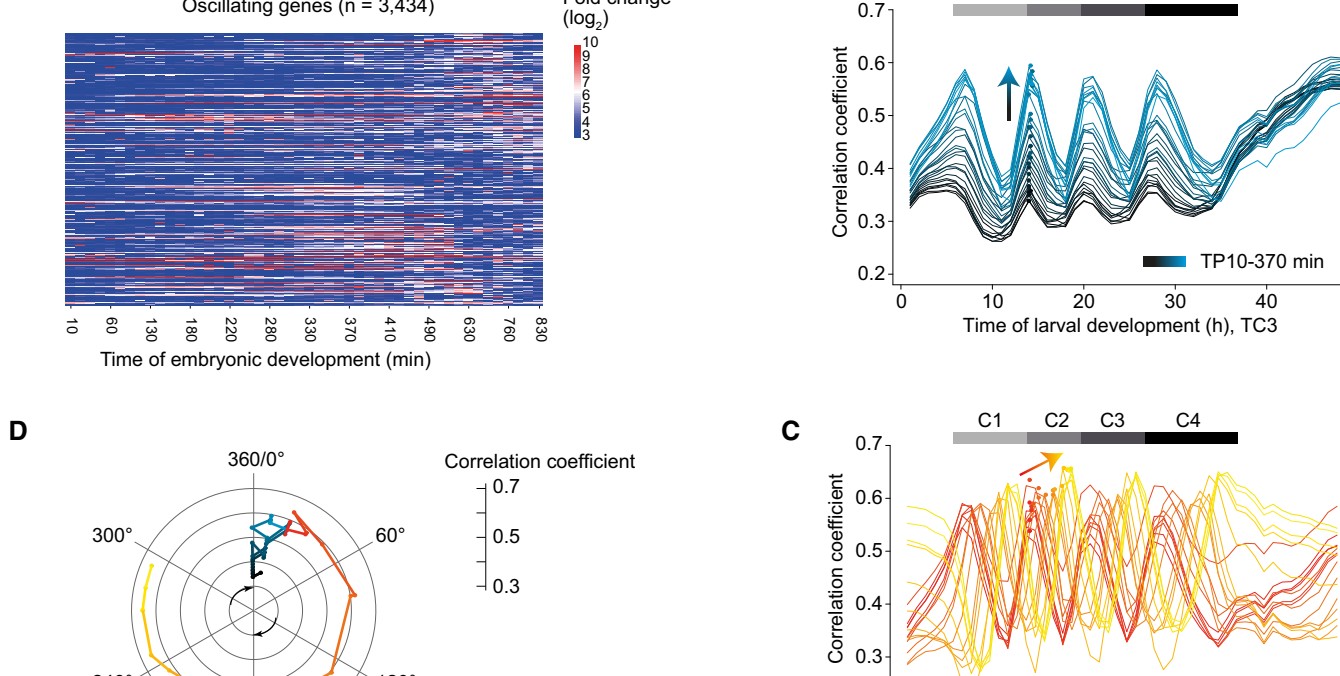

**Figure 6.  Transition to an oscillatory state during embryogenesis.**

A   Heatmap of log₂-transformed embryonic expression of oscillating genes, excluding L4 deviating genes, sorted by larval peak phase (defined in Fig 1).

B, C   Pairwise correlation coefficients between embryonic and larval time points (Fig EV5) plotted over larval time for embryonic TP10-370 min (B, *black-blue gradient*) and TP380-830 min (C, *red-yellow gradient*), respectively. Dots represent peaks of the correlation lines after spline analysis in the second oscillation cycle (C2), and arrows indicate trends. Horizontal gray bars indicate oscillation cycles C1 through C4 as in Fig 1C.

D   Polar plot of correlation coefficient peak over the time point in the second larval oscillation cycle (C2) at which the correlation peak is detected. TP14 is defined as 0° and correlates most highly to TP20, thus defined as 360°. Color scheme as in B and C.

Data information: All correlations were determined by Pearson correlation.

oscillating genes (Fig EV5A). When we plotted the correlation coefficients for each embryonic time point over larval time, we observed two distinct behaviors (Fig 6B and C), which separated at ~380 min (95%-CI: 317.6–444.2 min) (Fig EV5B and C): First, from the start of embryogenesis until ~380 min, the peak of correlation occurred always for the same larval time point, but the extent of correlation increased rapidly (Fig 6B). Second, past ~380 min of embryonic development, the peaks of correlation moved progressively from TP14 (which we define as 0°/360° because it demarcates the end of the first and the beginning of the second oscillation cycle in the fused time course; Fig 1C) toward TP19 (accordingly defined as 300°), but the extent of correlation increased only modestly (Fig 6C).

We conclude that the system adopts two distinct states during embryogenesis (Fig 6D): Initially, it approaches the oscillatory regime through increasing similarity to the oscillator phase TP14/0°. After completion of gastrulation and around the beginning of morphogenesis/organogenesis (Hall *et al*, 2017), it transitions into the oscillatory state and reaches, at hatching, a phase corresponding

to larval ~TP19/300°, where oscillations arrest until resumption later in L1.

## A shared oscillator phase for experimentally induced and naturally occurring bifurcations

The arrested states of the oscillator in both early L1 stage larvae and in adults are highly similar and resemble the oscillator state at TP19/300°. Therefore, we wondered whether state transitions of the system in response to changes in the developmental trajectory occurred also through this phase in other situations. To test this, we examined animals that exited from dauer arrest, a diapause stage that animals enter during larval development under conditions of environmental stress such as heat, crowding, or food shortage. Using a published time course of animals released from dauer arrest after starvation (Hendriks *et al*, 2014), we found that their expression patterns of oscillating genes correlated highly with those of animals initiating oscillations (TC3) in the L1 stage (Fig 7A and B). Additionally, gene expression patterns at 1 h through 5 h and at

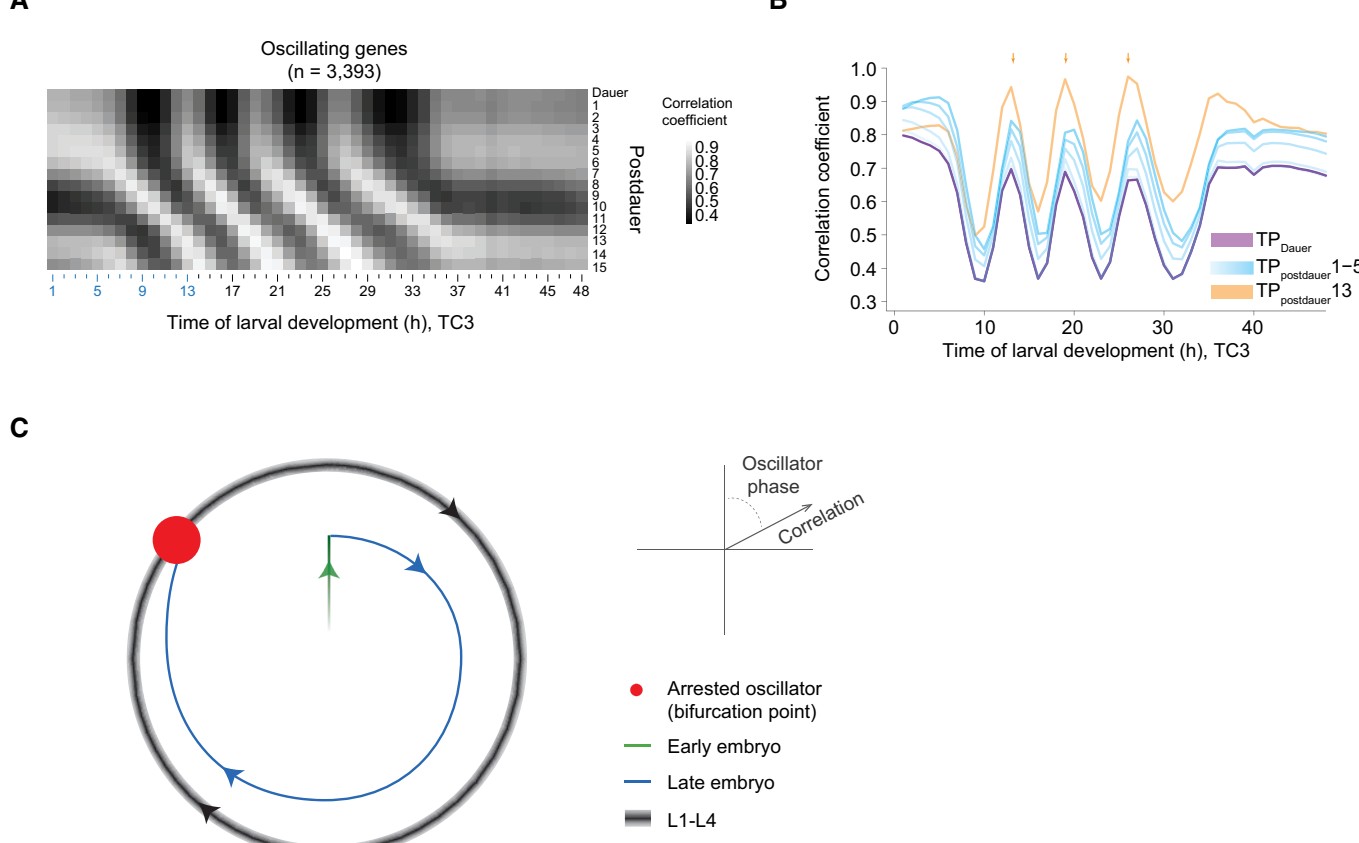

**Figure 7.  Re-initiation of oscillations after dauer from an arrested oscillator phase.**

A   Pairwise-correlation map of log₂-transformed oscillatory gene expression of dauer exit samples ("postdauer") and fused larval time course (TC3) samples.
B   Correlation of the indicated time points after plating dauer-arrested animals on food (TPpostdauer) to the fused larval time course, TC3. 'Dauer' indicates animals harvested before placing on food. Arrows indicate peaks of correlation to TP13/19/26 (300°) of TC3.
C   Schematic depiction of behavior of the *Caenorhabditis elegans* oscillator from embryo to adult. A phase-specific arrest (*red dot*) is observed at hatch, in early L1, young adults, and dauer-arrested animals. See Appendix Fig S7 for additional data supporting four similar oscillation cycles during L1 through L4.

13 h post-dauer were highly correlated with those of the repetitive TP13, TP19, and TP26 during continuous development. Hence, the system state during a period of quiescence during the first 5 h after placing animals on food resembles an arrest of the oscillation in a phase seen at TP19/300° of the continuous development time course.

We conclude that the system bifurcates in the same manner during continuous, unperturbed development, after hatching, and in response to a perturbation, namely starvation-induced dauer arrest.

# Discussion

In this study, we have characterized biological function and behavior of the *C. elegans* larval oscillator. Our results from single animal- and population-based analyses reveal a close coupling to development, and specifically molting, and imply that processes essential for molting may not be restricted to lethargus. We have observed that oscillations are highly similar during the four cycles (Fig 7C, Appendix Fig S7). Yet, oscillations cease and (re-) initiate several times during physiological development, and similar state transitions of the system can be induced through an external perturbation (Fig 7C). In particular, all non-oscillatory states correspond to an arrest of the oscillator in one specific phase. Hence, the observed bifurcations provide a conceptual model of how a developmental checkpoint can operate to halt larval development at a particular, repetitive point of development. Moreover, they constrain possible system architectures and properties as well as the choice and parametrization of mathematical models that can represent the system.

## A developmental oscillator with functions in and beyond molting and lethargus

The physiological function of the *C. elegans* oscillator has remained unclear. Here, we have demonstrated that it is coupled to molting. We propose that a function of the oscillator as a developmental clock provides a parsimonious explanation, but other models remain possible; e.g., the oscillator may facilitate an efficient molting process by anticipating the time of peak demand for cuticular building blocks and other factors.

Conventionally, molting is subdivided into three distinct steps, namely apolysis (severing of connections between the cuticle and the underlying epidermis), new cuticle synthesis, and ecdysis (cuticle shedding) (Lažetić & Fay, 2017). The first two occur during, and the latter terminates, lethargus, a period of behavioral quiescence. Evidently, the *C. elegans* oscillator imposes a temporal structure of gene expression that extends far beyond lethargus, with a majority of oscillating genes exhibiting gene expression peaks outside lethargus. In part, this may be explained by a role of the oscillator in coordinating other physiological processes with the molt (Ruaud & Bessereau, 2006).

However, molting itself also appears to comprise processes that occur outside lethargus. Specifically, we observed initiation of oscillations in embryos (which execute cuticle synthesis but neither apolysis nor ecdysis) at ~380 min into embryo development and thus long before the first signs of cuticle synthesis at ~600 min (Sulston *et al*, 1983). Instead, this time coincides with

formation of an apical extracellular matrix (ECM), the embryonic sheath. Four genes encoding components of this ECM, namely *sym-1; fbn-1; noah-1; and noah-2* (Vuong-Brender *et al*, 2017), are also required for larval molting or proper cuticle formation (Frand *et al*, 2005; Niwa *et al*, 2009), and we find that their expression oscillates with high amplitudes and peaks long before lethargus. Hence, ECM remodeling, and possibly other processes, crucial for molting can occur long before the onset of lethargus.

## Oscillatory state transitions and developmental checkpoints

We have observed a loss of oscillations under three distinct conditions, in early L1 stage larvae, dauer arrested animals, and adults. The similarity of the oscillator states under all three conditions is striking and involves an arrest in the same specific phase.

Formally, for the L1 arrest, we cannot distinguish between perturbation-induced or naturally occurring arrest, as the sequencing experiments required animal synchronization by hatching animals in the absence of food, causing a transient arrest of development. However, the fact that the L1 stage is extended also in animals hatched into food (Figs EV2C–E and 3B) suggests that they may adopt a similar arrested state even in the presence of food, perhaps because the nutritional resources in the egg (i.e., egg yolk) have become depleted by the time that hatching occurs. In other words, synchronization of L1 animals by hatching them in the absence of food may propagate a pre-existing transient developmental and oscillator arrest.

Irrespective of this interpretation, a key feature of the arrests that we observe under different conditions is that they always occur in the same phase. This is a behavior one would predict for a repetitive developmental checkpoint. Such a checkpoint has indeed been found to operate shortly after each larval molt exit, arresting development in response to a lack of food (Schindler *et al*, 2014). Importantly, developmental arrest does not result from an acute shortage of resources. Rather, it is a genetically encoded, presumably adaptive, response to nutritionally poor conditions, critically dependent on *daf-2*/IGFR signaling (Baugh, 2013; Schindler *et al*, 2014).

Within the limits of our resolution, the phase of the arrested oscillator corresponds to the phase seen around ecdysis. Thus, oscillations and development are synchronously arrested, and we propose that signals related to food sensing, metabolism, or nutritional state of the animal help to control the state of the oscillatory system and thereby developmental progression. An oscillator operating near a SNIC bifurcation appears ideally suited to processing such information, because it acts as a signal integrator; i.e., it becomes active when a signal threshold is surpassed (Izhikevich, 2000; Forger, 2017). This contrasts with the behavior of oscillators operating near a supercritical Hopf bifurcation, which function as resonators; i.e., they respond most strongly to an incoming signal of a preferred frequency. Hence, both the phase-specific arrest and the integrator function as characteristics of an oscillator operating in the vicinity of a SNIC bifurcation are physiologically relevant features of this *C. elegans* oscillator.

We note that checkpoints of the cell cycle have also been interpreted as bifurcations (Tyson *et al*, 2001, 2002). In this system, bifurcations separate stable $G_1$ and $S–G_2$ states from one another as

well as from an oscillatory M-phase state. This latter checkpoint in particular has been reported to involve a supercritical Hopf (Qu et al, 2003) or a SNIC bifurcation (Csikász-Nagy et al, 2006; referred to as SNIPER in this report). Although further conceptual and mechanistic similarities between the cell cycle checkpoints and the checkpoints of the C. elegans oscillator and development remain to be explored, this parallel suggests that implementation of checkpoints through system bifurcations may be a unifying concept in biology.

### Insights into oscillator architecture and constraints for mathematical modeling

The behavior of the oscillator that we characterized here constrains its architecture. Specifically, the change in period without a noticeable change in amplitude seen in L4 stage larvae is considered incompatible with the function of a simple negative feedback loop but compatible with the operation of interlinked positive and negative feedback loops (Tsai et al, 2008; Mönke et al, 2017). Indeed, among synthetic genetic oscillators, operation near a SNIC bifurcation is rare and seen only for so-called amplified negative feedback oscillators, which rely on interlinked negative and positive feedback loops (Purcell et al, 2010)—and only within a certain parameter space (Guantes & Poyatos, 2006; Conrad et al, 2008). Hence, our findings constrain not only possible oscillator architectures and mathematical models thereof, but also their parametrization.

Finally, we note that mathematical models of somitogenesis clocks, inspired by mechanistic knowledge about the identity of individual oscillator components and their wiring, tend to represent oscillators operating near a supercritical Hopf bifurcation (Jensen et al, 2010; Webb et al, 2016). This appears consistent with observations on isolated cells in vitro (Webb et al, 2016). At the same time, changes in period, observed along with changes in amplitude in zebrafish embryos during somite formation and prior to cessation of oscillation (Shih et al, 2015), are not compatible with a supercritical Hopf bifurcation. Thus, and because an analysis of bifurcation behavior of somitogenesis clocks in vivo is challenging due to a complex space dependence of oscillation features (Soroldoni et al, 2014), it remains to be answered whether and to what extent the C. elegans oscillator and the somitogenesis clocks share specific properties. In any case, a comparison of the similarities and differences in behaviors, architectures, and topologies will help to reveal whether and to what extent diverse developmental oscillators follow common design principles.

# Materials and Methods

### Caenorhabditis elegans strains

The Bristol N2 strain was used as wild type. The following transgenic strains were used:
HW1370: EG6699; xeSi136[F11E6.3p::gfp::h2b::pest::unc-54 3′UTR; unc-119 +] II (this study).
HW1939: EG6699; xeSi296[eft-3p::luc::gfp::unc-54 3′UTR, unc-119 (+)] II (this study).

HW2523: EG6699; xeSi437[qua-1p::gfp::h2b::pest::unc-54 3′UTR; unc-119 +] II (this study).
HW2526: EG6699; xeSi440[dpy-9p::gfp::h2b::pest::unc-54 3′UTR; unc-119 +] II (this study).

PE255: feIs5[sur-5p::luc::gfp; rol-6(su1006)] X (Lagido et al, 2008).
PE254: feIs5[sur-5p::luc::gfp; rol-6(su1006)] V (Lagido et al, 2008).

All transcriptional reporters and luciferase constructs produced for this study were generated using Gibson assembly (Gibson et al, 2009) and the destination vector pCFJ150 (Frøkjaer-Jensen et al, 2008). First, a starting plasmid was generated by combining NotI digested pCFJ150, with either Nhe-1::GFP-Pest-H2B or Nhe-1::luciferase::GFP (adapted from pSLGCV (Lagido et al, 2008)) and ordered as codon optimized, intron-containing gBlocks® Gene Fragment (Integrated DNA Technologies), and unc-54 3′UTR (amplified from genomic DNA) to yield pYPH0.14 and pMM001, respectively. Second, promoters consisting of either 2 kb upstream of the ATG or up to the next gene were amplified from C. elegans genomic DNA before inserting them into NheI-digested pYPH0.14 or pMM001. PCR primers and resulting plasmids are listed in the Appendix Table S2. Third, we obtained transgenic worms by single-copy integration into EG8079 worms, containing the universal ttTi5605 locus on chromosome II by following the published protocol for injection with low DNA concentration (Frøkjær-Jensen et al, 2012). All MosSCI strains were backcrossed at least twice.

### Method luciferase assay

Gravid adults were bleached, and single embryos were transferred by pipetting into a well of a white, flat-bottom, 384-well plate (Berthold Technologies, 32505). Embryos hatched and developed in 90 μl volume containing E. coli OP50 ($OD_{600} = 0.9$) diluted in S-Basal medium (Stiernagle, 2006), and 100 μM Firefly D-Luciferin (p.j.k., 102111). Plates were sealed with Breathe Easier sealing membrane (Diversified Biotech, BERM-2000). Luminescence was measured using a Luminometer (Berthold Technologies, Centro XS3 LB 960) for 0.5 s every 10 min for 72 h at 20°C in a temperature-controlled incubator and is given in arbitrary units.

Luminescence data were analyzed using an automated algorithm for molt detection on trend-corrected data as described previously (Olmedo et al, 2015), but implemented in MATLAB, and with the option to manually annotate molts in a Graphical User Interface. The hatch was identified as the first data point (starting from time point 4 to avoid edge effects) that exceeds the following value: the mean + 5*stdev of the raw luminescence of the first 20 time points. To quantify the duration of the molts, we subtracted the time point at molt entry from the time point at molt exit. To quantify the duration of larval stages, we subtracted the time point at molt exit of the previous stage (or time point at hatch for L1) from the time point at molt exit of the current stage. The duration of the intermolt was quantified as duration of the molt subtracted from duration of the larval stage. For statistical analysis, we assumed the durations to be normally distributed and used Welch two-sample and two-sided t-test, i.e., the function "t.test" of the package "stats" (version 3.5.1) (R Core Team) in R.

### RNA sequencing

For RNA sequencing, synchronized L1 worms, obtained by hatching eggs in the absence of food, were cultured at 25°C and collected hourly from 1 h until 15 h of larval development, or 5 h until 48 h of larval development, for L1–L2 time course (TC1) and L1–YA time course (TC2), respectively. A replicate experiment was performed at room temperature from 1 h until 24 h (TC4). RNA was extracted in Tri Reagent and DNase-treated as described previously (Hendriks *et al*, 2014). For TC2 and TC4, libraries were prepared using the TruSeq Illumina mRNA-seq (stranded—high input), followed by the Hiseq50 Cycle Single-end reads protocol on HiSeq2500. For TC1, libraries were prepared using the Illumina TruSeq mRNA-Seq Sample Prep Kit (Strand-sequenced: any), followed by the Hiseq50 Cycle Single-end reads protocol on a HiSeq2500.

### Processing of RNA-seq data

RNA-seq data were mapped to the *C. elegans* genome using the qAlign function (splicedAlignment = TRUE) from the QuasR package (Au *et al*, 2010; Gaidatzis *et al*, 2015) in R. Gene expression was quantified using qCount function from the QuasR package in R. For TC2 and Dauer exit (Hendriks *et al*, 2014) time courses, QuasR version 1.8.4 was used, and data were aligned to the ce10 genome using Rbowtie aligner version 1.8.0. For TC1, QuasR version 1.2.2 was used, and data were aligned to the ce6 genome using Rbowtie aligner version 1.2.0. For TC4 (Fig 2), RNA-seq data were mapped to the *C. elegans* ce10 genome using STAR with default parameters (version 2.7.0f) and reads were counted using htseq-count (version = 0.11.2).

Counts were scaled by total mapped library size for each sample. A pseudocount of 8 was added, and counts were $\log_2$-transformed. For TC2, lowly expressed genes were excluded (maximum $\log_2$-transformed gene expression - ($\log_2$(gene width)-mean($\log_2$(gene width))) $\leq 6$). This step was omitted in the early time courses because many genes start robust expressing only after 5–6 h. Expression data of the dauer exit time course were obtained from (Hendriks *et al*, 2014).

### Classification of genes by Cosine fitting

To classify genes, we applied cosine fitting to the $\log_2$-transformed gene expression levels from $t = 10$ h until $t = 25$ h of developmental time (mid-L1 until late L3) of TC2, when the oscillation period is most stable (Fig 4C). During this time, the oscillation period is approximately 7 h, which we used as fixed period for the cosine fitting. We built a linear model as described (Hendriks *et al*, 2014) using cos(ωt) and –sin(ωt) as regressors (with 13 degrees of freedom). In short, a cosine curve can be represented as follows:

$$C \cdot \cos(\omega t + \varphi) = A \cdot \cos(\omega t) - B \cdot \sin(\omega t)$$

with $A = C \cdot \cos (\varphi)$
and $B = C \cdot \sin (\varphi)$

From the linear regression ("lm" function of the package "stats" in R), we obtained the coefficients A and B, and their standard errors. A and B represent the phase and the amplitude of the oscillation:

$$phase = \arctan(A, B)$$

$$amplitude = \sqrt{A^2 + B^2}$$

As the density of the genes strongly decreased around 0.5 (Fig EV1C), we used $\log_2$(amplitude) $\geq 0.5$ as a first classifier. We propagated the standard error of the coefficients A and B to the amplitude using Taylor expansion in the "propagate" function (expr = expression(sqrt(((A^2)+(B^2))), ntype = "stat", do.sim = FALSE, alpha = 0.01) from the package "propagate" (version 1.0-6) (Spiess, 2018) in R. We obtained a 99% confidence interval (99%-CI) for each gene. As 99%-CI that does not include 0 is significant (*P*-value = 0.01), we used the lower boundary (0.5%) of the CI as a second classifier. Thus, we classified genes with an amplitude $\geq 0.5$ and lower CI-boundary $\geq 0$ as "oscillating" and genes with an < 0.5 or a lower CI-boundary < 0 were classified as "not oscillating" (Figs 1B and EV1C, Dataset EV1). Every gene thus has an amplitude and a peak phase (Dataset EV1). A peak phase of 0° is arbitrarily chosen, and thus, current peak phases are expected to differ systematically from the previously assigned peak phases (Hendriks *et al*, 2014). To compare the peak phases of TC2 with those of the previously published L3-YA time course (TC6), we calculated the phase difference (TC2–TC6) (Fig EV1D and E). We added 360° to the difference and used the modulus operator (% % 360), to maintain the circularity within the data. The coefficient of determination, $R^2$, was calculated by 1-(SSres/SStot), in which the SStot (total sum of squares) is the sum of squares in peak phase of the L1-YA time course. SSres (response sum of squares) is the sum of squares of the phase difference.

Our previous work (Hendriks *et al*, 2014) identified ~2,700 oscillating (i.e., rhythmically expressing) genes, a number that we now increase to 3,739 genes (24% of total expressed genes). We attribute this increase to a combination of slightly different cut-offs and a focus, in the new analysis, on the L1, L2 and L3 stages, where a constant oscillation period of ~7 h of facilitates cosine wave fitting. This contrasts with the situation in the previous experiment, which used data from the L3 and L4 stages and thus, as we reveal here, a time of changing period.

Even our current estimate is conservative, i.e., the "non-oscillating" genes contain genes that exhibit oscillatory expression with low amplitude or, potentially, strongly non-sinusoidal shapes. It is possible that such dynamics may play important roles for specific genes and processes and our data provide a resource to identify these in the future. However, here we focused on genes with robust and extensive oscillations to facilitate functional dissection of the oscillator.

### Classification of oscillating genes by Meta2D

As an alternative approach, we classified oscillating genes using the MetaCycle package (version 1.2.0) in R (Wu *et al*, 2016, 2019), which is an algorithm that incorporates three different algorithms, i.e., ARSER, JTK_CYCLE, and Lomb-Scargle, to detect periodic signals from time-series experiments. Similar to cosine fitting, we used the log2-transformed gene expression levels from $t = 10$ h until $t = 25$ h of developmental time (mid L1 until late L3) of TC2. We applied the meta2d algorithm (cycMethod = c("ARS","JTK")) with a period ranging between 4 and 9 h (minper = 4, maxperiod = 9,

ARSdefaultPer = 7), weighted scores based on the *P*-value of each method to calculate the integrated period length and phase (weightedPerPha = TRUE), and otherwise default parameters. [As the package documentation highlights a poor performance of the Lomb-Scargle algorithm in quantifying the oscillation amplitude, which was used as one of our cut-offs, we excluded it.] We classified genes with an amplitude ≥ 0.5 and FDR < 0.05 as "Meta2D" oscillating genes and compared them to the set of oscillating genes determined by cosine fitting.

**Time course fusion**

In order to obtain an RNA-seq time course spanning the complete larval development, we fused the L1–L2 time course (TC1, TP1–TP15) with the L1–YA course (TC2, TP5–TP48). To decide which time points to choose from the individual time courses, we correlated the gene expression of all genes ($n$ = 19,934) of both time courses against each other using the $\log_2$-transformed data with a pseudocount of 8 with Pearson correlation. In general, we saw good correlation between the two time courses, e.g., $TP5_{(TC1, L1–L2)}$ correlated well with $TP5_{(TC2, L1–YA)}$, etc. (Fig EV1B). We fused the two time courses at TP13, i.e., combined $TP1–TP13_{(TC1, L1–L2)}$ with $TP14–TP48_{(TC2, L1–YA)}$.

**Exclusion of genes based on L4 mean expression**

Given that oscillating genes were identified based on gene expression in TP10-TP25, when oscillation period is most stable, some genes showed deviating behavior in the last oscillation cycle, C4. Hence, for quantification of oscillation amplitude, period, and correlation, we excluded those genes. We determined the mean expression levels for each gene over time in oscillation cycles C2 (TP14-TP20), C3 (TP20-TP27), and C4 (TP27-TP36). Genes ($n$ = 291) were excluded if the absolute value of the difference in mean expression between L2 and L4 normalized by their mean difference exceeded 0.25, i.e.: abs((L2mean-Expr-L4meanExpr)/(0.5*(L2meanExpr+L4meanExpr)))>0.25.

**Quantification of oscillation amplitude**

To quantify the oscillation amplitude for each larval stage, we split the TC2 in 4 separate cycles, roughly corresponding to the developmental stages, i.e., C1: TP6-TP14, C2: TP14-TP20, C3: TP20-TP27, and C4: TP27-TP36 developmental time. We applied cosine fitting to C2, C3, and C4 as described above to the expression of oscillating genes in TC2, excluding genes with deviating mean expression in L4 as described above. We excluded C1, because amplitudes were sometimes difficult to call reliably. We used a fixed period of 7 h for C2-C3 and 8.5 h for C4 as determined by quantification of the oscillation period (Fig 4C). We applied a linear regression using the function "lm" of the package "stats" in R to find the relationship between the amplitudes across different stages, i.e., the slope. The correlation coefficient, r, was determined using the "cor" function (method = Pearson) of the package "stats" in R.

**Quantification of oscillation period**

For a temporally resolved quantification of the oscillation period, we filtered the mean-normalized $\log_2$-transformed gene expression levels of oscillating genes, excluding L4 deviating genes (we selected TP5-TP39, because oscillations cease at ~TP36 and the inclusion of 3 additional time points avoided edge effects) using a Butterworth filter ("bwfilter" function of the package "seewave" (version 2.1.0) (Sueur *et al*, 2008) in R, to remove noise and trend-correct the data. The following command was used to perform the filtering: bwfilter(-data, f = 1, $n$ = 1, from = 0.1, to = 0.2, bandpass = TRUE, listen = FALSE, output = "matrix"). The bandpass frequency from 0.1 to 0.2 (corresponding to 10- and 5-h period, respectively) was selected based on the Fourier spectrum obtained after Fourier transform ("fft" function with standard parameters of the package "stats"). As an input for the Hilbert transform, we used the Butterworth-filtered gene expression. The "ifreq" function (with standard parameters from the package "seewave") was used to calculate the instantaneous phase and frequency based on the Hilbert transform (see Appendix). To determine the phase progression over time, we unwrapped the instantaneous phase (ranging from 0 to $2\pi$ for each oscillation) using the "unwrap" function of the package "EMD" (version 1.5.7) (Kim & Oh, 2018) in R. To avoid edge effects, we removed the first 4 data points (TP5-TP8) and last 3 data points (TPTP37-TP39) of the unwrapped phase (retaining TP9-TP36). The angular velocity is defined as the rate of phase change, which we calculated by taking the derivative of the unwrapped phase. The instantaneous period was determined by $2\pi$/angular velocity and was plotted for each gene individually and as mean in a density plot. The mean of the instantaneous period over all oscillating genes was used to reconstruct a "global" oscillation by taking the following command: sin(cumsum(mean angular velocity)) and plotted together with a 7-h period oscillation and the mean normalized expression of a representative gene, *col-147*.

**Correlation analyses of RNA-seq data**

$\log_2$-transformed data were filtered for oscillating genes and then plotted in a correlation matrix using the R command cor(data, method = "pearson"). The correlation line plots represent the correlations of selected time points to the fused full developmental time course (Fig EV4) and are specified in the line plot.

To reveal the highest correlations for a selected time point, we analyzed the correlation line of this time point between TP7 and TP36 (the time in which oscillations occur) using a spline analysis from Scipy (v1.2.1) (Jones *et al*, 2001) in python ("from scipy.inter-polate import InterpolatedUnivariateSpline" with k = 4) and stored the spline as variable "spline". We identified peaks of the correlation line by finding the zeros of the derivative of the spline (cr_-points = spline.derivative().roots()). The highest correlations of the respective correlation line were thus the value of the spline at the time point where the spline derivative was zero and the value was above the mean of the correlation line (cr_vals = spline(cr_pts) followed by pos_index = np.argwhere(cr_vals>np.mean(data.iloc [i])) and peak_val = cr_vals[pos_index]). Thus, we identified the correlation of particular time points (e.g., TP14–TP19) with their corresponding time points in the next oscillation cycle. Thereby, we were able to identify cycle time points as described in the results section. We defined the first cycle time point, e.g., TP14 of cycle 2, as 0°, and the last unique one, TP19, as 300°. TP14 (0° of cycle 2) is also 360° of cycle 1. Note that a sampling interval of 1 h means that a TP in one cycle may correlate equally well to two adjacent TPs in

another cycle, as seen for instance in the correlation of TP13 to TP26 and TP27. The spline interpolation places the peak of correlation in the middle of these time points at ~TP26.5. The spline analysis thus annotates cycle points correctly even in C4 which has an extended period.

We performed correlation analyses without mean normalization of expression data, and hence, correlation values cannot be negative but remain between 0 and 1. We made this decision because a correlation analysis using mean-centered data, where correlations can vary between -1 and +1, requires specific assumptions on which time points to include or exclude for mean normalization, and because it is sensitive to gene expression trends. However, we confirmed, as a proof of principle, the expected negative correlation of time points that are in antiphase when using mean-centered data (Appendix Fig S8) using oscillating genes without a trend in L4 in TC3 ($n$ = 3,393).

### GO-term analysis

GO-term analysis was performed using the GO biological process complete option (GO ontology database, release 2019-02-02) from the online tool PANTHER (PANTHER Classification System) (over-representation test, release 2019-03-08, standard settings).

### Tissue-specific analysis

In order to reveal whether particular tissues are enriched in oscillating genes, we used single cell sequencing data from Cao *et al* (2017). In particular, we used Appendix Table S6: Differential expression test results for the identification of tissue-enriched genes where each gene's highest and second highest tissue expression and the ratio thereof is reported. We selected tissue-specific genes based on a ratio > 5 and a qvalue < 0.05 (these criteria reduced the number of genes to investigate). Using this list of genes, we calculated the percentage of tissues present in all genes and in oscillating genes, respectively, using the function "Counter" from "collections" in python (*labels, values = zip (\*Counter(tissue_info_thr["max.tissue"]).items())*). In order to obtain the enrichment of tissues, we divided the percentage of tissue X among oscillating genes in the tissue enriched dataset by the percentage of tissue X among all genes in the tissue-enriched dataset and plotted the resulting values. The list of tissue-specific oscillating genes was further used to investigate the peak phases within one tissue by plotting a density plot of the peak phase (from Fig 1) for every tissue. As we lack data below 0 degree and above 360 degree, density values at these borders are distorted as the density is calculated over a moving window. Since we are confronted with cyclical data, and thus, 0 degree corresponds to 360 degree, we added and subtracted 360 degree to each phase value, thus creating data that ranged from -360 degree to 720 degree which allowed us to plot the correct density at the borders 0 and 360 degree. We used python (pandas, v0.24.1) to plot these data using the following command:

data_tissue ["Phase"].plot(kind = "kde", linewidth = 5, alpha = 0.5, bw = 0.1).

### Identification of first gene expression peaks in L1 larvae

To identify the first peak of oscillating genes, we used a spline analysis from Scipy (v1.2.1) (Jones *et al*, 2001) in Python ("from scipy.interpolate import InterpolatedUnivariateSpline") from TP3 to TP13. We chose these time points to remove false positives in the beginning due to slightly higher noise for the first 2 time points as well as not to identify the second peak which occurred at $\geq$ TP14 for some very early genes. The function used was "InterpolatedUnivariateSpline" with k = 4. After constructing the spline, we identified the zeros of the derivative and chose the time point value with the highest expression value and a zero derivative as the first peak time point.

### Embryonic gene expression time course

Embryonic gene expression data were obtained from Hashimshony *et al* (2015) and represented precisely staged single embryos at 10-min intervals from the 4-cell stage up to muscle movement and every 10–70 min thereafter until 830 min. We obtained the gene count data from the Gene Expression Omnibus database under the accession number GSE50548, for which sequencing reads were mapped to WBCel215 genome and counted against WS230 annotation.

We normalized the gene counts to the total mapped library size per sample, added a pseudocount of 8, and log$_2$-transformed the data. We selected genes according to the larval oscillating gene annotation, with L4 deviating genes excluded, and plotted their embryonic expression patterns according to peak phase in larvae. The embryonic time course was correlated with the fused larval time course (TC3) using the "cor" function (method = "pearson") of the package "stats" in R (Fig EV5A). Correlation line plots were generated by plotting the correlation coefficients for each embryonic time point over larval time. To identify the peaks of the correlation lines with a resolution higher than the sampling frequency, we interpolated the correlation lines using the "spline" function ($n$ = 240, method = 'fmm') of the package "stats" in R. To call the peaks of the interpolated correlation lines, we applied the "findpeaks" function (with nups = 5, ndowns = 5) of the package "pracma" (version 2.2.5) on the time points on the interpolated time points 10–185, that cover the four cycles. To find the embryonic time point at which oscillations initiate, we plotted the larval TP in cycle 2 at which the correlation peak occurred over embryonic time (Fig EV5B) and determined the intersection of the two linear fits, using the "solve" function of the package "Matrix" (version 1.2-17) (Bates & Maechler, 2018) and the "lm" function of the package "stats" in R, respectively. To determine the 95%-CI of the x-coordinate of the intersect, the standard error of the slope a and the intercept b of the two linear fits was propagated using Taylor expansion in the "propagate" function (expr = expression((b1-b2)/(a2-a1)), ntype = "stat",do.sim = FALSE, alpha = 0.05) from the package "propagate" (version 1.0-6) in R. The pairwise correlation map was generated with the "aheatmap" function of the package "NMF" (version 0.21.0) (Gaujoux & Seoighe, 2010), and the 3D plot was generated with the "3Dscatter" function of the package "plot3D" (version 1.3) (Soetaert, 2017) in R.

### Dauer exit gene expression time course

The dauer exit time course TP1-15 were obtained from Hendriks *et al* (2014), https://www.ncbi.nlm.nih.gov/geo/query/acc.cgi? acc = GSE52910. TP0 is from the same experiment and is accessible

through GEO accession number GSM4448413 (https://www.ncbi.nlm.nih.gov/geo/query/acc.cgi?acc = GSM4448413).

## Time-lapse imaging of single animals

Single worm imaging was done by adapting a previous protocol (Turek *et al*, 2015) and is similar to the method reported in Gritti *et al* (2016). Specifically, we replaced the previous 3.5-cm dishes with a "sandwich-like" system: The bottom consisted of a glass cover slip onto which two silicone isolators (GRACE Bio-Labs, SKU: 666103) with a hole in the middle were placed on top of each other and glued onto the glass cover slip. We then placed single eggs inside the single OP50 containing chambers, which were made of 4.5 % agarose in S-basal. The chambers including worms were then flipped 180 degree and placed onto the glass cover slip with the silicone isolators, so that worms faced the cover slip. Low melt agarose (3 % in S-basal) was used to seal the agarose with the chambers to prevent drying out or drifts of the agarose chambers during imaging. The sandwich-like system was then covered with a glass slide on the top of the silicone isolators to close the system.

We used a 2× sCMOS camera model (T2) CSU_W1 Yokogawa microscope with 20× air objective, NA = 0.8 in combination with a 50-μm disk unit to obtain images of single worms. For a high throughput, we motorized the stage positioning and the exchange between confocal and brightfield. We used a red LED light to combine brightfield with fluorescence without closing the shutter. Additionally, we used a motorized z-drive with 2 μm step size and 23 images per z-stack. The 488 nm laser power for GFP imaging was set to 70 %, and a binning of 2 was used.

To facilitate detection of transgene expression and oscillation, we generated reporters using the promoters of genes that exhibited high transcript levels and amplitudes, and where GFP was concentrated in the nucleus and destabilized through fusion to PEST::H2B (see strain list above). We placed embryos into chambers containing food (concentrated bacteria HT115 with L4440 vector) and imaged every worm with a z-stack in time intervals of 10 min during larval development in a room kept at ~21°C, using a double camera setting to acquire brightfield images in parallel with the fluorescent images. We exploited the availability of matching fluorescent and brightfield images to identify worms by machine learning. After identification, we flattened the worm at each time point to a single-pixel line and stacked all time points from left to right, resulting in one kymograph image per worm. We then plotted background-subtracted GFP intensity values from the time of hatch (*t* = 0 h), which we identified by visual inspection of the brightfield images as the first time point when the worm exited the egg shell.

Time-lapse images were analyzed using a customized KNIME workflow (see Data availability). We analyzed every worm over time using the same algorithm. First, we identified the brightest focal planes per time point by calculating the mean intensity from all focal planes per time point and selecting the focal planes that had a higher intensity than the mean. Then, we maximum-projected the GFP images over Z per time point and blurred the DIC image and also max projected over Z (blurring the DIC improved the machine learning process later on). All images per worm over time were analyzed by Ilastik machine learning in order to identify the worm in the image. The probability map from Ilastik

was used to select a threshold that selected worms of a particular experiment best. (The threshold might change slightly as DIC images can look slightly different due to differences in the sample prep amongst experiments.) Using a customized ImageJ plug-in, we straightened the worm. The straightened GFP worm image was then max projected over Y which resulted in a single-pixel line representing the GFP intensities in a worm and after stacking up all the single-pixel lines in Y direction, and we obtained the kymographs. In order to remove noise coming from the head and tail regions of the worm due to inaccuracy of the machine learning, we measured mean GFP intensities per time point ranging from 20 % until 80 % of the worms anterior–posterior axis. For background subtraction, we exploited the fact that only the nuclei were GFP positive and thus subtracted the minimum intensity value between GFP nuclei from their intensity values.

After the KNIME workflow, we imported the measured GFP intensities into Python and analyzed the traces using a Butterworth filter and Hilbert transform analysis (both from Scipy, v1.2.1 (Jones *et al*, 2001)). We used the Butterworth bandpass filter using b, a = butter(order = 1, [low,high], btype = "band") with low = 1/14 and high = 1/5, corresponding to 14- and 5-h periods, respectively. We then filtered using filtfilt(b, a, data, padtype = 'constant') to linearly filter backwards and forwards.

For individual time points where the worm could not be identified by the Ilastik machine learning algorithm, we linearly interpolated (using interpolation from pandas, v0.24.1, (McKinney, 2010)) using "pandas.series.interpolate(method = 'linear', axis = 0, limit = 60, limit_direction = 'backward'", between the neighboring time points to obtain a continuous time series needed for the Hilbert transform analysis. Using Hilbert transform, we extracted the phase of the oscillating traces for each time point and specifically investigated the phase at molt entry and molt exit for our different reporter strains.

In order to determine time points in which worms are in lethargus, we investigated pumping behavior. As the z-stack of an individual time point gives a short representation of a moving worm, it is possible to determine whether animals pump (feeding, corresponds to intermolt) or not (lethargus/molt). Additionally to the pumping behavior, we used two further requirements that needed to be true in order to assign the lethargus time span: First, worms needed to be quiescent (not moving, and straight line), and second, a cuticle needed to be shed at the end of lethargus. Usually worms start pumping one to two time points before they shed the cuticle. This analysis was done manually with the software ImageJ, and results were recorded in an excel file, where for every time point, the worms' behavior was denoted as 1 for pumping and as 0 for non-pumping.

To determine a possible connection between oscillations and development, we applied error propagation, assuming normal distribution of the measured phases and larval stage durations. Thereby, we exploited the inherent variation of the oscillation periods and developmental rates among worms, rather than experimental perturbation, to probe for such a connection. The durations are represented with the mean ($\mu$) and the standard deviation ($\sigma^2$). We define the phase at either molt entry or molt exit as

$$\theta_{\text{entry}} \equiv \frac{2\pi}{T_o} \cdot T_{IM} \sim (\mu, \sigma^2)$$

and

$$\theta_{\text{exit}} \equiv \frac{2\pi}{T_o} \cdot T_L \sim (\mu, \sigma^2)$$

the intermolt duration and $T_L \sim (\mu_L, \sigma_L^2)$ the larval stage duration of the respective larval stages. These calculations result in a phase with mean $\mu$ and a standard deviation $\sigma$ at molt entry and molt exit, respectively, for each larval stage indicated. Should the two processes be coupled as in scenario 2, we would expect $\sigma_{observed} < \sigma_{calculated}$.

To calculate the phase at molt entry and molt exit with error propagation, we used the "uncertainties" package (v3.0.3) (Lebigot) in python. The larval stage duration and intermolt duration and period were treated as ufloat numbers, representing the mean and standard deviation of the distributions coming from our measurement (e.g., $7.5 \pm 0.2$). These distributions were then used to calculate the expected phase at molt entry (using the intermolt duration) and molt exit (using the larval stage duration) using the above-mentioned formulas. This resulted in the phase being represented by an ufloat number and thus a distribution which we used for plotting after normalizing for the mean to compare the variation of the data. In order to confirm that the package worked correctly, we performed the error propagation ourselves using the formula:

$$\sigma_{\text{entry error propagated}} = 2\pi \cdot \frac{T_{IM}}{T_o} \sqrt{\left(\frac{\sigma T_{IM}}{T_{IM}}\right)^2 + \left(\frac{\sigma T_o}{T_o}\right)^2}$$

and

$$\sigma_{\text{exit error propagated}} = 2\pi \cdot \frac{T_L}{T_o} \sqrt{\left(\frac{\sigma T_L}{T_L}\right)^2 + \left(\frac{\sigma T_o}{T_o}\right)^2}$$

which led to the same results as the package.

### Correlation analysis of phase and developmental events

Using single worm imaging data, we compared the absolute time at which we observed a specific but arbitrarily chosen unwrapped phase from the GFP oscillation with the absolute time at which we observed either molt entry or molt exit.

The unwrapped phases we chose were 11rad for L2 comparisons and 18rad for L3 comparisons. We chose these phases because they occurred late in L2 and L3, respectively. The scatterplot reveals a good correlation with Pearson correlation coefficients exceeding 0.9 which was calculated using the pandas (v0.24.1) function df.corr (method = "pearson"). We used linear models to fit the data with the function "regression.linear_model.OLS" from statsmodels.api (v0.10.1) assuming an intercept of 0. From these models, we obtained the slope with 95% confidence intervals. The predicted values from the linear model are plotted in blue with the shaded area corresponding to the 95% confidence intervals.

### Simulations

To examine the bifurcation dynamics in response to temporally changing parameters, we simulated the model

$$\frac{dx}{dt} = x(\beta - x^2 - y^2) - 2\pi y(1 - \lambda y),$$

$$\frac{dy}{dt} = y(\beta - x^2 - y^2) + 2\pi x(1 - \lambda y), \quad \forall x, y \in \mathbb{R}$$

where $x$ and $y$ are two variables describing the state of the oscillator, and $\beta$ and $\lambda$ are the Hopf and SNIC parameters, respectively. Default values for $\beta$ and $\lambda$ were 1 and 0, respectively. The model was integrated using the ODE solver in the Scipy package (v1.3.1) (Jones *et al*, 2001) in python ("from scipy.integrate import odeint"). Stochastic simulations were performed by using the Euler–Maruyama method.

The model can be better understood when the system is transformed into polar coordinates, i.e.,

$$\frac{dr}{dt} = r(\beta - r^2),$$

$$\frac{d\theta}{dt} = 2\pi(1 - \lambda r \sin \theta), \quad \forall r, \theta \in \mathbb{R}^+ \tag{1}$$

where $r = \sqrt{x^2 + y^2}$ and $\theta = \tan^{-1}(y/x)$ are the amplitude and phase of the system. For fixed parameter values, this system shows oscillations for positive values of $\beta$ and $|\lambda r| < 1$, with maximum amplitude given by the radius of the limit cycle, $r_{LC} = \sqrt{\beta}$. At the limit cycle, for a fixed $\beta$, the period of the oscillator is given by

$$T = \frac{1}{\sqrt{1 - \beta \lambda^2}}$$

### Hopf simulations

To simulate the effects of a Hopf bifurcation under a slowly changing parameter, Equation 1 was simulated with a fixed value of $\lambda = 0$ and

$$\beta(t) = k_\beta t,$$

where $k_\beta$ is the rate of change of $\beta$. For illustrative purposes, the value of $\beta(t)$ was defined to be 1 when $\beta(t) > 1$. All deterministic simulations for a slowly changing $\beta$ were performed with initial conditions of $r = 10^{-5}$ and stochastic simulations with a value of $r_0 = 0$. The initial phase was defined to be $\theta_0 = \pi/2$.

Solutions for the amplitude go through an interval where the solution remains close to the steady state and then jumps suddenly to a neighborhood of the limit cycle. However, the amplitude approaches asymptotically the limit cycle, and thus, the system was determined to have reached the limit cycle if the difference between the rate of change of the radius of the limit cycle and the amplitude was sufficiently small, i.e.,

$$\left| \frac{dr}{dt} - k_\beta \right| < \varphi.$$

For the simulations, the threshold $\varphi = 0.01$.

## SNIC simulations

To simulate the effects of a SNIC bifurcation under a slowly changing parameter, Equation 1 was simulated for a value of $\beta = 1$ and

$$\lambda(t) = 1 - k_\lambda t,$$

where $k_\lambda$ is the rate of change of $\lambda$. As the effect for $\lambda$ is symmetric, values were constrained to the positive real numbers including zero. Negative values were set to zero.

The system was initialized at the SNIC bifurcation point on the limit cycle; i.e., the initial conditions for phase and amplitude were defined to be $\theta_0 = \pi/2$ and $r_0 = 1$, respectively.

## Data availability

The datasets and computer code produced in this study are available in the following databases:

- All RNA-seq data: NCBI's Gene Expression Omnibus (Edgar *et al*, 2002) SuperSeries accession number GSE133576 (https://www.ncbi.nlm.nih.gov/geo/query/acc.cgi?acc = GSE133576).
- Single worm imaging analysis code: Github (https://github.com/fmi-basel/ggrosshans_SWIanalysis).
- Single worm imaging KNIME workflow: Nodepit (https://nodepit.com/workflow/com.nodepit.space%2Fyannickhauser%2Fpublic%2FWorm%20images_final_with%20workaround.knwf), including a FMI-specific plug-in (fmi-ij2-plugins, https://doi.org/10.5281/zenodo.3560533).
- Simulations of bifurcations: https://github.com/fmi-basel/ggrosshans_BifurcationModel_Meeuse2020

Expanded View for this article is available online.

## Acknowledgements

We thank Stephane Thiry, Kirsten Jacobeit, and the FMI Functional Genomics Facility for RNA sequencing; Iskra Katic for help in generating transgenic strains; Maria Olmédo and Henrik Bringmann for introducing us to the luciferase and the single animal imaging assays, respectively, Dimos Gaidatzis and Michael Stadler for advice on computational analyses; Laurent Gelman for help with imaging; and Benjamin Towbin, Prisca Liberali, and Luca Giorgetti for comments on the manuscript. M.W.M.M. is a recipient of a Boehringer Ingelheim Fonds PhD fellowship. This work is part of a project that has received funding from the European Research Council (ERC) under the European Union's Horizon 2020 research and innovation program (Grant agreement No. 741269, to H.G.). The FMI is core-funded by the Novartis Research Foundation.

## Author contributions

G-JH and YPH performed RNA sequencing time courses. MWMM and YPH analyzed RNA sequencing data. MWMM performed and analyzed luciferase assays. LJMM performed simulations. GB developed the graphical user interface for the luciferase data. YPH acquired and analyzed single worm imaging data. JE wrote the KNIME workflow for the single worm imaging. CT conceived parts of the analysis. HG, MWMM, and YPH conceived the project and wrote the manuscript.

## Conflict of interest

The authors declare that they have no conflict of interest.

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
