## [Review Process File · Molecular Systems Biology]

Developmental function and state transitions of a gene expression oscillator in *C. elegans*

Milou Meeuse, Yannick Hauser, Lucas Morales Moya, Gert-Jan Hendriks, Jan Eglinger, Guy Bogaarts, Charisios Tsiairis, and Helge Großhans

DOI: [10.15252/msb.20209498](https://doi.org/10.15252/msb.20209498)

Corresponding author(s): Helge Großhans (helge.grosshans@fmi.ch)

Review Timeline:

Submission Date:	4th Feb 20
Editorial Decision:	27th Feb 20
Revision Received:	8th May 20
Editorial Decision:	5th Jun 20
Revision Received:	15th Jun 20
Accepted:	22nd Jun 20

Editor: Jingyi Hou

Transaction Report:

27th Feb 2020

Manuscript Number: MSB-20-9498

Title: Developmental function and state transitions of a gene expression oscillator in *C. elegans*

Author: Milou Meeuse

Yannick Hause

Gert-Jan Hendriks

Jan Eglinger

Guy Bogaarts

Charisios Tsiaris

Helge Großhans

Thank you for submitting your work to Molecular Systems Biology. We have now heard back from the three reviewers who agreed to evaluate your study. As you will see below, the reviewers acknowledge that the presented findings seem potentially interesting. They raise however a series of concerns, which we would ask you to address in a major revision.

I think that the reviewers' recommendations are rather clear and there is therefore no need to repeat the comments listed below. In particular, additional analyses are required to provide certain levels of mechanistic insights, as commented by reviewer #3.

All other issues raised by the reviewers need to be satisfactorily addressed as well. As you may already know, our editorial policy allows in principle a single round of major revision and it is therefore essential to provide responses to the reviewers' comments that are as complete as possible. Please feel free to contact me in case you would like to discuss in further detail any of the issues raised by the reviewers.

On a more editorial level, please do the following:

- Please provide a .docx formatted version of the manuscript text (including legends for main figures, EV figures and tables). Please make sure that the changes are highlighted to be clearly visible.
- Please provide individual production quality figure files as .eps, .tif, .jpg (one file per figure).
- Please provide a .docx formatted letter INCLUDING the reviewers' reports and your detailed point-by-point responses to their comments. As part of the EMBO Press transparent editorial process, the point-by-point response is part of the Review Process File (RPF), which will be published alongside your paper.
- Please note that all corresponding authors are required to supply an ORCID ID for their name upon submission of a revised manuscript.
- We have replaced Supplementary Information by the Expanded View (EV format). In this case, due to the large number of additional Figures, all of them they can be included in a PDF now called

Appendix. Appendix figures (and Tables) should be labeled and called out as: "Appendix Figure S1, Appendix Figure S2... Appendix Table S1..." etc. Each legend should be below the corresponding Figure/Table in the Appendix. Please include a Table of Contents in the beginning of the Appendix. For detailed instructions regarding expanded view please refer to our Author Guidelines: (<https://www.embopress.org/page/journal/17444292/authorguide#expandedview>).

- Before submitting your revision, primary datasets (and computer code, where appropriate) produced in this study need to be deposited in an appropriate public database (see <https://www.embopress.org/page/journal/17444292/authorguide#dataavailability>). - Dataset #1
- Dataset #2>

The accession numbers and database should be listed in a formal "Data Availability " section (placed after Materials & Method) that follows the model below (see also <https://www.embopress.org/page/journal/17444292/authorguide#dataavailability>). Please note that the Data Availability Section is restricted to new primary data that are part of this study.

Data availability

- We would encourage you to include the source data for figure panels that show essential quantitative information. Additional information on source data and instruction on how to label the files are available at < <https://www.embopress.org/page/journal/17444292/authorguide#sourcedata> >.

- All Materials and Methods need to be described in the main text. We would encourage you to use 'Structured Methods', our new Materials and Methods format. According to this format, the Material and Methods section should include a Reagents and Tools Table (listing key reagents, experimental models, software and relevant equipment and including their sources and relevant identifiers) followed by a Methods and Protocols section in which we encourage the authors to describe their methods using a step-by-step protocol format with bullet points, to facilitate the adoption of the methodologies across labs. More information on how to adhere to this format as well as downloadable templates (.doc or .xls) for the Reagents and Tools Table can be found in our author guidelines: < <https://www.embopress.org/page/journal/17444292/authorguide#researcharticleguide> >. An example of a Method paper with Structured Methods can be found here: .

- Please provide a "standfirst text" summarizing the study in one or two sentences (approximately 250 characters, including space), three to four "bullet points" highlighting the main findings and a "synopsis image" (550px width and max 400px height, jpeg format) to highlight the paper on our homepage.

- When you resubmit your manuscript, please download our CHECKLIST (http://embopress.org/sites/default/files/Resources/EP_Author_Checklist.xls) and include the completed form in your submission. *Please note* that the Author Checklist will be published alongside the paper as part of the transparent process <http://msb.embopress.org/authorguide#transparentprocess>.

If you feel you can satisfactorily deal with these points and those listed by the referees, you may wish to submit a revised version of your manuscript. Please attach a covering letter giving details of the way in which you have handled each of the points raised by the referees. A revised manuscript will be once again subject to review and you probably understand that we can give you no guarantee at this stage that the eventual outcome will be favorable.

Yours sincerely,

Jingyi Hou
Editor
Molecular Systems Biology

If you do choose to resubmit, please click on the link below to submit the revision online *within 90 days*.

Link Not Available

IMPORTANT: When you send your revision, we will require the following items:

1. the manuscript text in LaTeX, RTF or MS Word format
2. a letter with a detailed description of the changes made in response to the referees. Please specify clearly the exact places in the text (pages and paragraphs) where each change has been made in response to each specific comment given
3. three to four 'bullet points' highlighting the main findings of your study
4. a short 'blurb' text summarizing in two sentences the study (max. 250 characters)
5. a 'thumbnail image' (550px width and max 400px height, Illustrator, PowerPoint or jpeg format), which can be used as 'visual title' for the synopsis section of your paper.
6. Please include an author contributions statement after the Acknowledgements section (see <https://www.embopress.org/page/journal/17444292/authorguide>)
7. Please complete the CHECKLIST available at (<http://bit.ly/EMBOPressAuthorChecklist>). Please note that the Author Checklist will be published alongside the paper as part of the transparent process (<https://www.embopress.org/page/journal/17444292/authorguide#transparentprocess>).
8. Please note that corresponding authors are required to supply an ORCID ID for their name upon submission of a revised manuscript (EMBO Press signed a joint statement to encourage ORCID adoption). (<https://www.embopress.org/page/journal/17444292/authorguide#editorialprocess>)

Currently, our records indicate that the ORCID for your account is 0000-0002-8169-6905.

Link Not Available

The system will prompt you to fill in your funding and payment information. This will allow Wiley to send you a quote for the article processing charge (APC) in case of acceptance. This quote takes into account any reduction or fee waivers that you may be eligible for. Authors do not need to pay any fees before their manuscript is accepted and transferred to the publisher.

REFeree REPORTS

Reviewer #1:

Summary

In this article, the authors use RNA sequencing of synchronized *C. elegans* larvae to study oscillatory gene expression during post-embryonic development. They combine separate RNA-seq time courses to create the first temporal gene expression map for all 48h hours of larval development. Most prominently they show that when the molting cycle oscillations arrest, either permanently (upon entry into adulthood) or temporarily (in early-L1, or when recovering from dauer), gene expression appears 'fixed' at a particular phase of the molting cycle oscillation. In addition, the authors use time-lapse imaging of individual animals to show that entry into and exit from molts occurs at stereotypical phase of the oscillation, and are likely functionally interlinked. They observe lengthened periods of molting cycle oscillations and larval stage duration in larval stages where the molting cycle oscillator is either 'starting up' (L1) or 'winding down' (L4). Together, these observations point to a specific mechanism for generating and controlling the molting cycle oscillations: a Saddle Node on an Invariant Cycle (SNIC) bifurcation.

General remarks

How timing of development is regulated remains largely a mystery and the molting cycle oscillations observed in *C. elegans* are a potentially an important mechanism to explain some aspects of timing of *C. elegans* development. However, the mechanism that generates these oscillations remains unknown. In addition, it is not clear how these oscillations stop once adulthood is reached. A general challenge is that molting cycle oscillations impact many genes, while what constitutes the core oscillator is not known. Therefore, the use of whole-genome techniques such as RNA-seq, as the authors do, appears crucial for advancing.

Even though this manuscript does not address what the core components of the molting cycle oscillator are, they use quantitative analysis of RNA-seq data in an innovative way to constrain potential mechanisms. In particular, their observation that the oscillations arrest at a specific phase

of the oscillation points strongly to a SNIC bifurcation as the underlying mechanism, which represents a significant conceptual advance in our understanding of this process. Therefore, I can recommend publication if some of my more detailed remarks (see below) are addressed.

Major points

- p. 9, line 16 - p. 10, line 9.

The authors use error propagation to present evidence that molting cycle oscillations are functionally linked to the process of molting itself. I find the explanation of how this analysis works, both in Fig. 1F and in the methods, difficult to follow and hence it was hard for me to judge the validity of the results in Fig. 1G.

I think I probably understand the overall idea: if *qua-1* oscillations and e.g. molt entry are not linked, any variability in their timing will be uncorrelated. In that case, you could have a large oscillation period T_O , but a short intermolt duration T_d , and vice versa, leading to a large variation in the phase at molt entry. If on the other hand they are linked, then variability in timing will be correlated: large T_O then also implies large T_d , meaning that in the equation for the these deviations from the mean cancel each other out and the error in phase will be small.

However, I find the explanation in the Methods of how these errors are calculated confusing at a number of points:

The methods section has a single equation for both the phase at molt entry and molt exit: should there not be two different equations for that, as molt entry and exit happen at different phases of the *qua-1* oscillation?

The data in Fig. 3G seems to be based on the single-animal *qua-1* time-lapse measurements, but it is not clear to me how that data is connected to the quantities, i.e. T_O , in the formula for the phase. Is it just the mean and SD in the period of *qua-1* oscillations that is used? If so, is it the period of the oscillation only in the larval stage considered, e.g. L2, or is it a measurement that is somehow averaged over all of development? Would it not be possible to measure the phase at molt entry or exit directly in each single animal, by comparing time-lapse *qua-1* expression dynamics to the RNA-seq data?

It is not clear to me how errors in T_O and T_d are propagated in practice. Are the means and SDs for T_O and T_d first computed from single animal data, and then inserted into the equation for $\theta = 2\pi/T_O * T_d$ to somehow generate mean and SD for θ ? Here, I am confused by the fact that there are two differently formulated equations for θ , that in practice appear identical, so it is not clear to me how they differ exactly.

What also makes things more complex that necessary is that a Python package is used to propagate errors, which makes the exact process followed opaque. In particular, for a simple division $A = B/C$ where B and C vary independently, like the equation for θ , the propagated error is $dA = |A| \sqrt{(dB/B)^2 + (dC/C)^2}$. Is there a reason that I am missing why this simple equation does not apply in the case considered here? If not, I think using the formula above makes the process easier to understand.

In addition, I wonder whether there are other/additional ways to show more directly that variability in timing of *qua-1* oscillations and molt entry/exit are correlated. Would it be possible to directly plot

the time at which qua-1 expression reaches a particular phase (for instance the population-averaged phase at which molt entry is observed to occur) against the time of molt entry. If these two events vary independently, one would expect a point cloud with poor correlation. If they covary, these points should cluster around a line with high correlation.

In conclusion, I think it is likely that the results and conclusions in Fig. 3G are correct, but I think this conclusion could be strengthened by additional analysis and the process to reach these conclusions should be more clearly written down.

Minor points

- p.14, line 18-23. "... this oscillator phase more conducive to state transitions."

In this section, the authors say that they test whether the arrested state with at TP19 is more conducive to state transitions, which I assume mean transitions from oscillatory to non-oscillatory. Is this really what the experiment tests? I read this to mean that whenever the worm wants to start or stop oscillations it arranges its gene expression to resemble that at TP19. However, that suggests that gene expression during dauer is different from that at TP19, and only starts to resemble it upon exposure to food. However, gene expression during dauer was, as far as I saw, not examined. Could it be a possibility that whenever larval development arrests, gene expression remains like TP19 until it resumes again, i.e. dauer animals always show TP19-like expression?

- p. 21, line 2-4. "...contrasts with changes in both amplitude and period..."

This sentence was not clear to me, does it refer to the fact that for supH bifurcations only amplitude but not period changes?

- p.20, line 22 - p. 21, line 10

In the discussion, and also a bit in the introduction, the molting cycle oscillator is linked and compared to the somitogenesis clock. Whereas I can see that there are some superficial similarities (that it is a developmental process and oscillations start en stop) otherwise the overlap between the two systems does not seem so strong to me. In my mind, cell cycle oscillations seem a much better comparison, as they control timing of cellular processes and can also start and stop at will via checkpoints. SNIC-type bifurcations have been examined in the context of cell cycle oscillations. For that reason, it seems more important to me to discuss whether molting cycle oscillations share properties with the cell cycle rather than with the somitogenesis clock.

- Fig. 2D. I don't understand this figure. Is the x-axis the time of the first peak in L1? It was also not clear what the key message of this panel is: that genes that peak at the same time in L1 typically have similar phases during L2-L4? The main text only says that Fig. 2D shows that occurrence of first peaks was 'globally well correlated', but that is too vague for me.

- Fig. 2E. I don't understand the point of this panel: does it only show that peak time in L1 is the same from experiment to experiment? If so, that observation doesn't seem worth a panel in a main figure, but rather in the SI.

- Fig. 3B. It is not clear what is shown in the figures. Do I see individual lines, representing a single animal, that are colored blue during intermolt and red during molt? A more complete caption would be helpful in this respect (and also in general for the other figures).

- Fig. 3F. As I discussed above, I don't understand this figure. Apart from the larger point clouds on the two circles, I see no difference between the coupled and not coupled sub-panels.
- Fig. 5B. It is not clear from the text, caption or Fig. S5, how this figure is different from Fig. 1A. It is not only the removal of the strange L4 genes, correct? In any case, this deserves to be explained better.
- Fig. 5C,D. I cannot see the gray lines very well in these panels. I am also confused by the caption. Is it correct that all lines shown are 'lines of correlation', i.e. how closely gene expression at that time represents that at the TP examined, but they are colored differently for the different TPs? Now, the caption somehow suggests that TP37-48 are special. Also, is TP19 special compared to TP13 and TP26/27, or just used as an example? Also why does the article talk about TP26/27, but not e.g. TP12/13? From the lines in these panels it is not so clear why for TP26 the maximum is more poorly defined than for TP13.
- Fig. 6B,C. It is hard to see the colorbar in the legends for Tp10-370 (B) and TP 380-830 (C). It might be better to stretch it in the horizontal direction.
- Fig. 6D,E. I find Fig. 6D confusing, and there is not much help in the text or the caption to distill the key message. Fig. 6E shows the phase information that is also implicitly shown already in 6B,C. To me it seems that these panels could be removed without much impact on the paper.

Reviewer #2:

In parallel to other groups Helge Grosshans discovered a few years ago a global transcriptional oscillator in the early development of worms. In addition to the circadian clock and the somite oscillator this system represents a genetic oscillator of immediate physiological relevance (in contrast to the poorly understood role of other famous oscillators such as glycolysis, p53, NFkB). Thus a deeper understanding can provide insights to developmental processes and the physiological role of rhythms. Unfortunately, the driving gene-regulatory networks are not known at this stage. Consequently, conceptual models are appropriate as in the early days of chronobiology (Aschoff, Wever, Pittendrigh, Daan, Winfree...). In the submitted manuscript very carefully analyzed data are related to advanced bifurcation theory (subcritical Hopf versus SNIC). The convincing evidence for a SNIC bifurcation implies the coexistence of negative and positive feedback loops. Thus future models can be constrained by the precisely documented periods, phases, and amplitudes characteristic for SNIC bifurcations. Biologically, the coupling of the oscillations to developmental checkpoints is quite interesting.

Specific comments:

1. Introduction: ... changes in the state of the oscillator system (or bifurcations)... The authors should be more careful in the distinction of bifurcations and transitions due to slowly varying conditions. Bifurcations are defined by topological changes of dynamical systems due to parameter variations. This implies that transients are not considered. If external parameters are varied very slowly sudden changes can be observed reflecting bifurcations of the underlying dynamical system. However, transients cannot be excluded completely. An application of this concept are bifurcations and chaos in voice signals where parameters such as pitch or loudness vary much slower than vocal fold vibrations.

2. I suggest to mention at the beginning of the results the number of expression profiles, replicates, and the sampling time.
3. There are better ways to write $2^{0.5}$.
4. page 7, line 18: remove "a"
5. page 10, lines 16,17 Here again bifurcations are defined heuristically. A comparison of true bifurcations and sudden transitions due to slowly varying parameters could profit from simulations of generic examples. For example, the exact bifurcation of a Hopf normal form could be compared to the observed variations of periods and amplitudes for a slowly varying Hopf parameter. Then instead of a precise square root increase of the amplitude a more smooth onset of oscillations becomes visible. This could be contrasted to simulations at a SNIC bifurcation.
6. I suggest to write subcritical Hopf instead of supH.
7. Pearson correlation is sensitive to outliers. A brief comparison with Spearman correlation would be interesting.
8. page 39: Why 13 degrees of freedom? In my eyes, harmonic regression and trend elimination requires less parameters.
9. page 40, line 11: Underlined T?

Reviewer #3:

This is an engaging paper that builds on prior work by this group on the gene expression oscillations that occur during the various developmental stages of *C. elegans*.

The authors first extend their previous work on developmental oscillations in this system by assaying at all stages of development. This shows that oscillations occur at every stage of development. In this study, they were able to then "glue" the gene expression profiles from the larval stages together to form a comprehensive dataset that could be used to track 3680 oscillating transcripts through the whole time course. Because they also assayed gene expression before oscillations began, and as they terminated.

The authors then sought to explain the observed gene expression cycle behaviors by examining two potential models: a superficial Hopf bifurcation and a SNIC. Clearly, the oscillations fitted with a SNIC bifurcation, given that their amplitude remained high and period lengthened in the final oscillation prior to adulthood (and transition to non-oscillatory behavior).

In general, the paper is welcomed development of the group's previous findings, but there are some issues that should be addressed prior to publication, particularly with respect to some exploration of mechanism.

Major points

1. There is a lot of correlation in this paper, and although the SNIC model does move towards mechanism, the authors don't really talk about how / why oscillations begin/end. In particular, there is a latency between putting the worms on food and the oscillations starting. Is the starvation, followed by feeding, the cue that starts the oscillations (albeit with a phase delay)? Does gene ontology etc. reveal anything interesting in this regard? It would be much more informative if the authors try to address this by looking specifically into metabolic connections (which seem likely), e.g. in the context of Crosby et al. Cell 2019 (PMID: 31030999) and insulin signaling and its relationship to the circadian clockwork. Something similar might be involved here.

2. The classification of genes as rhythmic is based on a cosine fitting algorithm / linear modelling. The authors do not seem to have used robust statistics here (using confidence interval only as far as I can see). There is no mention of multiple-testing correction and false discovery rate estimations for the classifications. The authors need to carry out and state what cut-offs they have used, perhaps using more widely used algorithms for temporally dense datasets (e.g. metaCycle, Cyclops, JTK_Cycle, RAIN). These alternative classifications will yield more robust statistics and FDR estimates. Also, these algorithms should allow the authors to assess period ranges (e.g. 4-9 hour period genes) rather than constraining the fitting to a 7 hour period.

3. The discussion is way too long. It should be cut to a third of the size.

Minor points

1. The authors use Hilbert transforms on a few occasions. They should give a bit of background to non-specialists about why this was used and what it does to the data.

We thank the reviewers for their constructive comments on our manuscript, which we have comprehensively addressed as detailed in the following:

Reviewer #1:

Summary

In this article, the authors use RNA sequencing of synchronized *C. elegans* larvae to study oscillatory gene expression during post-embryonic development. They combine separate RNA-seq time courses to create the first temporal gene expression map for all 48h hours of larval development. Most prominently they show that when the molting cycle oscillations arrest, either permanently (upon entry into adulthood) or temporarily (in early-L1, or when recovering from dauer), gene expression appears 'fixed' at a particular phase of the molting cycle oscillation. In addition, the authors use time-lapse imaging of individual animals to show that entry into and exit from molts occurs at stereotypical phase of the oscillation, and are likely functionally interlinked. They observe lengthened periods of molting cycle oscillations and larval stage duration in larval stages where the molting cycle oscillator is either 'starting up' (L1) or 'winding down' (L4). Together, these observations point to a specific mechanism for generating and controlling the molting cycle oscillations: a Saddle Node on an Invariant Cycle (SNIC) bifurcation.

General remarks

How timing of development is regulated remains largely a mystery and the molting cycle oscillations observed in *C. elegans* are a potentially an important mechanism to explain some aspects of timing of *C. elegans* development. However, the mechanism that generates these oscillations remains unknown. In addition, it is not clear how these oscillations stop once adulthood is reached. A general challenge is that molting cycle oscillations impact many genes, while what constitutes the core oscillator is not known. Therefore, the use of whole-genome techniques such as RNA-seq, as the authors do, appears crucial for advancing.

Even though this manuscript does not address what the core components of the molting cycle oscillator are, they use quantitative analysis of RNA-seq data in an innovative way to constrain potential mechanisms. In particular, their observation that the oscillations arrest at a specific phase of the oscillation points strongly to a SNIC bifurcation as the underlying mechanism, which represents a significant conceptual advance in our understanding of this process. Therefore, I can recommend publication if some of my more detailed remarks (see below) are addressed.

We thank this reviewer for his/her favorable evaluation of our work and have addressed the points raised by him/her as detailed below.

Major points

- p. 9, line 16 - p. 10, line 9.

The authors use error propagation to present evidence that molting cycle oscillations are functionally linked to the process of molting itself. I find the explanation of how this analysis works, both in Fig. 1F and in the methods, difficult to follow and hence it was hard for me to judge the validity of the results in Fig. 1G.

I think I probably understand the overall idea: if qua-1 oscillations and e.g. molt entry are not linked, any variability in their timing will be uncorrelated. In that case, you could have a large oscillation period T_O , but a short intermolt duration T_d , and vice versa, leading to a large variation in the phase at molt entry. If on the other hand they are linked, then variability in timing will be correlated: large T_O then also implies large T_d , meaning that in the equation for these deviations from the mean cancel each other out and the error in phase will be small.

However, I find the explanation in the Methods of how these errors are calculated confusing at a number of points:

The methods section has a single equation for both the phase at molt entry and molt exit: should there not be two different equations for that, as molt entry and exit happen at different phases of the qua-1 oscillation?

The one equation we showed was generic and could be applied to either situation. However, we agree that this could be confusing and have therefore now added two individual equations, as suggested by the reviewer.

The data in Fig. 3G seems to be based on the single-animal qua-1 time-lapse measurements, but it is not clear to me how that data is connected to the quantities, i.e. T_O , in the formula for the phase. Is it just the mean and SD in the period of qua-1 oscillations that is used? If so, is it the period of the oscillation only in the larval stage considered, e.g. L2, or is it a measurement that is somehow averaged over all of development? Would it not be possible to measure the phase at molt entry or exit directly in each single animal, by comparing time-lapse qua-1 expression dynamics to the RNA-seq data?

We apologize for the lack of clarity. We have changed the text in the manuscript to explain our approach more clearly. All measurements and calculations performed in Fig. 3H (previously Fig. 3G) correspond to single worm imaging results using transcriptional reporters only. As we could not measure developmental events such as molting in our RNA-seq time courses, we lack the ability to accurately define phase at molt entry and exit.

Indeed, for T_O , we calculate the mean of the period and the standard deviation of the GFP oscillation of a reporter per larval stage. In addition, we calculate the means of the individual stage durations, either intermolt T_{IM} or larval stage duration T_L , and their standard deviations σ^2 . These standard deviations are then used to propagate the errors. We assumed a normal distribution of the measured phases and larval stage durations. Thereby, we exploited the inherent variation of the oscillation periods and developmental rates among worms, rather than experimental perturbation, to probe for such a connection. The durations are represented with the mean (μ) and the standard deviation (σ). We define the phase at either molt entry or molt exit as

$$\theta_{entry} \equiv \frac{2\pi}{T_O} * T_{IM} \sim (\mu, \sigma^2) \text{ and}$$

$$\theta_{exit} \equiv \frac{2\pi}{T_o} * T_L \sim (\mu, \sigma^2)$$

with $T_o \sim (\mu_o, \sigma_o^2)$ being the period of oscillation, $T_{IM} \sim (\mu_{IM}, \sigma_{IM}^2)$ the intermolt duration and $T_L \sim (\mu_L, \sigma_L^2)$ the larval stage duration of the respective larval stages. These calculations result in a phase with mean μ and an expected standard deviation σ at molt entry and molt exit respectively for each larval stage indicated. Should the two processes be coupled as in scenario 2, we would expect

$$\sigma_{observed} < \sigma_{expected}$$

It is not clear to me how errors in T_O and T_d are propagated in practice. Are the means and SDs for T_O and T_d first computed from single animal data, and then inserted into the equation for theta = 2p/T_O * T_d to somehow generate mean and SD for theta? Here, I am confused by the fact that there are two differently formulated equations for theta, that in practice appear identical, so it is not clear to me how they differ exactly.

What also makes things more complex that necessary is that a Python package is used to propagate errors, which makes the exact process followed opaque. In particular, for a simple division $A = B/C$ where B and C vary independently, like the equation for theta, the propagated error is $dA = |A| \sqrt{(dB/B)^2 + (dC/C)^2}$. Is there a reason that I am missing why this simple equation does not apply in the case considered here? If not, I think using the formula above makes the process easier to understand.

We appreciate the reviewer's comment, which we have addressed by comparing the results of the package "uncertainties" with those of error propagation by this formulae:

$$\sigma_{entry\ error\ propagated} = 2\pi * \frac{T_{IM}}{T_o} \sqrt{\left(\frac{\sigma_{T_{IM}}}{T_{IM}}\right)^2 + \left(\frac{\sigma_{T_o}}{T_o}\right)^2} \text{ and}$$

$$\sigma_{exit\ error\ propagated} = 2\pi * \frac{T_L}{T_o} \sqrt{\left(\frac{\sigma_{T_L}}{T_L}\right)^2 + \left(\frac{\sigma_{T_o}}{T_o}\right)^2},$$

(These calculations were performed for larval stages L2 and L3 independently.) The results were identical, which is now indicated in the Methods section.

In addition, I wonder whether there are other/additional ways to show more directly that variability in timing of qua-1 oscillations and molt entry/exit are correlated. Would it be possible to directly plot the time at which qua-1 expression reaches a particular phase (for instance the population-averaged phase at which molt entry is observed to occur) against the time of molt entry. If these two events vary independently, one would expect a point cloud with poor correlation. If they co-vary, these points should cluster around a line with high correlation.

We agree with the reviewer's expectation that if the two features co-vary, they should cluster around a line with high correlation, and this indeed what we observe; all correlations are above 0.9. However, this finding does not allow us to conclude that the features are coupled because it

is unclear what correlation we can expect from a scenario where development and gene expression oscillations are not coupled, but driven by independent, yet accurate oscillators. This is exactly the kind of information that the error propagation analysis provides, which we have therefore retained. However, we agree that the correlation analysis may help readers to get a better intuition for the situation and, therefore, have now included it as novel Fig 3F and Fig. S3. Moreover, and in response to this reviewer's separate comment below we have re-drawn the schematic Fig. 3G (previously Fig, 3F) to clarify the distinct expectations.

In conclusion, I think it is likely that the results and conclusions in Fig. 3G are correct, but I think this conclusion could be strengthened by additional analysis and the process to reach these conclusions should be more clearly written down.

Minor points

- p.14, line 18-23. "... this oscillator phase more conducive to state transitions."

In this section, the authors say that they test whether the arrested state with at TP19 is more conducive to state transitions, which I assume mean transitions from oscillatory to non-oscillatory. Is this really what the experiment tests? I read this to mean that whenever the worm wants to start or stop oscillations it arranges its gene expression to resemble that at TP19. However, that suggests that gene expression during dauer is different from that at TP19, and only starts to resemble it upon exposure to food. However, gene expression during dauer was, as far as I saw, not examined. Could it be a possibility that whenever larval development arrests, gene expression remains like TP19 until it resumes again, i.e. dauer animals always show TP19-like expression?

We apologize for our lack of clarity here. The reviewer is exactly right, we did not mean to imply that the worm "arranges" for its gene expression to resemble TP19, but that whenever larval development arrests, gene expression remains like TP19. This seems indeed consistent with the fact that also after oscillator offset, in adults, this expression pattern is largely retained. We have corrected this sentence accordingly. Indeed, we have further tested this assumption by including in our analysis a sequencing sample of dauer arrested animals (i.e., prior to placing them on food) (Fig. 7A, B). (We had previously sequenced the dauer sample but not included it in our analysis for the post-dauer time course to be more similar to the full developmental time course.) We find that it exhibits maximum correlation to the same larval time points as the samples that were collected during the first hours on plate.

- p. 21, line 2-4. "...contrasts with changes in both amplitude and period..."

This sentence was not clear to me, does it refer to the fact that for supH bifurcations only amplitude but not period changes?

Yes, the reviewer's assumption is correct. We have rephrased this conclusion to clarify this point.

- p.20, line 22 - p. 21, line 10

In the discussion, and also a bit in the introduction, the molting cycle oscillator is linked and compared to the somitogenesis clock. Whereas I can see that there are some superficial similarities (that it is a developmental process and oscillations start en stop) otherwise the overlap between the two systems does not seem so strong to me. In my mind, cell cycle oscillations seem a much better comparison, as they control timing of cellular processes and can also start and stop at will via checkpoints. SNIC-type bifurcations have been examined in the context of cell cycle oscillations. For that reason, it seems more important to me to discuss whether molting cycle oscillations share properties with the cell cycle rather than with the somitogenesis clock.

We agree that at this point, the level of overlap between the somitogenesis clock and the *C. elegans* oscillator is unclear. Nonetheless, we think that it is relevant to look at these in comparison to understand shared vs. unique feature of developmental oscillators.

We also thank the reviewer for pointing out the potential analogy to the cell cycle. Indeed, we are very intrigued by the potential parallels. We had initially been reluctant to discuss them because we were afraid of confusing the readers. Specifically, we wanted to avoid the impression of a direct, mechanistic connection between cell cycle arrest and developmental/oscillation arrest, which we consider unlikely given that most cells in *C. elegans* larvae are postmitotic. Yet more importantly, the term “oscillator” is often used rather loosely in connection with the cell cycle. This can be appropriate, but confusing in the context of our discussion, since the cell cycle does not normally (outside embryos) function as a limit cycle oscillator. Instead, it appears to involve a succession of bistable switches (reviewed in Tyson et al., Nat Rev Mol Cell Biol 2001, Bioessays 2002). That said, it does contain an oscillatory (mitotic) module, which may involve a supercritical Hopf (Qu et al., Biophys J 2003) or a SNIPER (i.e., SNIC; Csikasz-Nagy et al., Biophys J 2006) bifurcation as a checkpoint.

Given these considerations (and the request by reviewer #3 to keep the overall discussion section short), we have now addressed the reviewer’s request by adding the following short section: “We note that checkpoints of the cell cycle have also been interpreted as bifurcations (Tyson et al., Nat Rev Mol Cell Biol 2001, Bioessays 2002). In this system, bifurcations separate stable G_1 and $S-G_2$ states from one another as well as from an oscillatory M-phase state. This latter checkpoint in particular has been reported to involve a supercritical Hopf (Qu et al., Biophys J 2003) or a SNIC bifurcation (Csikasz-Nagy et al., Biophys J 2006; referred to as SNIPER in this report). Although further conceptual and mechanistic similarities between the cell cycle checkpoints and the checkpoints of the *C. elegans* oscillator and development remain to be explored, this parallel suggests that implementation of checkpoints through system bifurcations may be a unifying concept in biology.”

- Fig. 2D. I don't understand this figure. Is the x-axis the time of the first peak in L1? It was also not clear what the key message of this panel is: that genes that peak at the same time in L1 typically have similar phases during L2-L4? The main text only says that Fig. 2D shows that occurrence of first peaks was 'globally well correlated', but that is too vague for me.

Yes, the reviewer has read the figure correctly. We have revised the figure legend to clarify this point. The key message is that the oscillations are 'structured' from the beginning. This already provides a first hint that the non-oscillatory state corresponds to an arrest of the oscillator rather than a unique stable state of the system.

- Fig. 2E. I don't understand the point of this panel: does it only show that peak time in L1 is the same from experiment to experiment? If so, that observation doesn't seem worth a panel in a main figure, but rather in the SI.

Yes, the reviewer has read the figure correctly. Since we agree with the reviewer that subsequent analyses make this point quite clearly (namely by demonstrating a SNIC bifurcation, for which this is the expected behavior), we have decided to omit this replicate experiment altogether.

- Fig. 3B. It is not clear what is shown in the figures. Do I see individual lines, representing a single animal, that are colored blue during intermolt and red during molt? A more complete caption would be helpful in this respect (and also in general for the other figures).

Yes, the reviewer has read the figure correctly. To help other readers with this, we have now carefully reviewed this and all other captions/figure legends and revised them if necessary to provide relevant information more clearly.

- Fig. 3F. As I discussed above, I don't understand this figure. Apart from the larger point clouds on the two circles, I see no difference between the coupled and not coupled sub-panels.

We apologize for a lack of clarity in this schematic figure. The larger point clouds are indeed the relevant feature, they represent a larger phase spread in the case that molting and oscillations are not coupled. We have now revised this figure (now Fig. 3G) extensively to make this point more clearly, and we have also added further detail in the figure legend.

- Fig. 5B. It is not clear from the text, caption or Fig. S5, how this figure is different from Fig. 1A. It is not only the removal of the strange L4 genes, correct? In any case, this deserves to be explained better.

We apologize for not being sufficiently clear about this in the figure legends, which we have now revised to address this issue: Fig. 1A shows a pairwise correlation plot for all(!) genes, whereas Fig. 5B specifically looks at oscillating genes, as the relevant subgroup for this analysis. In Fig. S5 (now Fig. EV4), we replot this part from Fig. 5B to visualize the analysis that we perform, which we consider important to enable non-specialists to follow the analysis. At the same time, we sought to keep it separate from the main figure because this illustration appeared to give unnecessary emphasis to a single time point, TP19.

- Fig. 5C,D. I cannot see the gray lines very well in these panels. I am also confused by the caption. Is it correct that all lines shown are 'lines of correlation', i.e. how closely gene expression at that time represents that at the TP examined, but they are colored differently for the different TPs? Now, the caption somehow suggests that TP37-48 are special. Also, is TP19 special compared to TP13 and TP26/27, or just used as an example? Also why does the article

talk about TP26/27, but not e.g. TP12/13? From the lines in these panels it is not so clear why for TP26 the maximum is more poorly defined than for TP13.

We apologize for the confusion. Indeed, all lines are ‘lines of correlation’, and we showed them for all time points for the sake of completeness. However, some (those in gray) are irrelevant for the present analysis, hence we chose a design where they are close to background. We have now omitted them entirely in the revised figures to avoid further confusion.

TP37–48 are indeed the relevant (special) feature that we are investigating in this panel. TP19 is used as an example of what we also see with TP13 and TP26/27. We chose TP19 arbitrarily as the timepoint in the middle of the time course, thus being similarly distant to early L1 and young adult time points. The reviewer is also correct that we could safely refer to TP26 as the peak, even though, as a matter of course, our hourly sampling frequency means that sampling time points will typically not represent exactly the times of peak correlation. The notation TP26/27 was a left-over from an earlier, preliminary analysis, and we agree that it is unnecessarily complicated; so we have now changed it to TP26.

- Fig. 6B,C. It is hard to see the colorbar in the legends for Tp10-370 (B) and TP 380-830 (C). It might be better to stretch it in the horizontal direction.

We thank the reviewer for this helpful suggestion, which we have followed.

- Fig. 6D,E. I find Fig. 6D confusing, and there is not much help in the text or the caption to distill the key message. Fig. 6E shows the phase information that is also implicitly shown already in 6B,C. To me it seems that these panels could be removed without much impact on the paper.

We agree with the reviewer that the information shown in Fig. 6D,E is implicit in Fig. 6B,C, but felt that it was important to show this result explicitly, in part to make it more obvious even to less careful readers than this reviewer. As a compromise, we have now moved Fig. 6D to the Extended View material (Fig. EV5) but retained Fig. 6E (now Fig. 6D) as a main figure.

Reviewer #2:

In parallel to other groups Helge Grosshans discovered a few years ago a global transcriptional oscillator in the early development of worms. In addition to the circadian clock and the somite oscillator this system represents a genetic oscillator of immediate physiological relevance (in contrast to the poorly understood role of other famous oscillators such as glycolysis, p53, NFkB). Thus a deeper understanding can provide insights to developmental processes and the physiological role of rhythms. Unfortunately, the driving gene-regulatory networks are not known at this stage. Consequently, conceptual models are appropriate as in the early days of chronobiology (Aschoff, Wever, Pittendrigh, Daan, Winfree...). In the submitted manuscript very carefully analyzed data are related to advanced bifurcation theory (subcritical Hopf versus SNIC). The convincing evidence for a SNIC bifurcation implies the coexistence of negative and positive feedback loops. Thus future models can be constrained by the precisely documented periods, phases, and amplitudes characteristic for SNIC bifurcations. Biologically, the coupling

of the oscillations to developmental checkpoints is quite interesting.

We thank the reviewer for his/her positive review of our work and share his/her view on the importance of conceptual models at this early stage of research into the *C. elegans* oscillator.

Specific comments:

1. Introduction: ... changes in the state of the oscillator system (or bifurcations)... The authors should be more careful in the distinction of bifurcations and transitions due to slowly varying conditions. Bifurcations are defined by topological changes of dynamical systems due to parameter variations. This implies that transients are not considered. If external parameters are varied very slowly sudden changes can be observed reflecting bifurcations of the underlying dynamical system. However, transients cannot be excluded completely. An application of this concept are bifurcations and chaos in voice signals where parameters such as pitch or loudness vary much slower than vocal fold vibrations.

If we understand the reviewer correctly, s/he would like to point out a difference between ‘transition’ (an observable behavior) and ‘bifurcation’ (a change of state of the system that underlies the behavior). In this view, a transition is a manifestation of the bifurcation, and these two may not always occur at the same time. For instance, in the case of slowly varying parameters, the system may not respond immediately once the bifurcation point is reached, but do so only when the parameter has reached some distance from the bifurcation point. Thus, there may be a delay between the experimentally observed transition and the underlying bifurcation, the dynamics of the observed transition may differ from that of the bifurcation of the underlying system, and/or, before settling into the new state, the system may adopt a transient state. This point is well taken, and we now mention the difference between bifurcation and observed transition in the Introduction (p. 3). As requested in the reviewer’s point #5 below, we also address this point through simulations.

2. I suggest to mention at the beginning of the results the number of expression profiles, replicates, and the sampling time.

This information was already provided in Fig. S1. To emphasize it more, we have now made this an Expanded View item (Fig. EV1) so that it will be displayed prominently in the online version of the article.

3. There are better ways to write $2^{0.5}$.

We have been unable to find any instance where we have used this writing. We did use $2^{0.5}$ once, which seems an appropriate and accepted way of writing this number. At any rate, we have now changed it to $\log_2(\text{amplitude}) \geq 0.5$.

4. page 7, line 18: remove "a"

We have adjusted this. Thank you.

5. page 10, lines 16,17 Here again bifurcations are defined heuristically. A comparison of true bifurcations and sudden transitions due to slowly varying parameters could profit from simulations of generic examples. For example, the exact bifurcation of a Hopf normal form could be compared to the observed variations of periods and amplitudes for a slowly varying Hopf parameter. Than instead of a precise square root increase of the amplitude a more smooth onset of oscillations becomes visible. This could be contrasted to simulations at a SNIC bifurcation.

This point, as we understand, repeats and extends this reviewer's comment #1. Specifically, the reviewer appears to wonder whether in a delayed supercritical Hopf bifurcation, period changes might occur for the observable transition (in addition to the canonical amplitude changes), and whether observed amplitude changes might differ from those that occur with an instantaneous parameter change.

We can rule out that a delayed supercritical Hopf bifurcation explains the period extension that we observe prior to oscillation offset in adults, because in a Hopf normal form, the bifurcation parameter affects only the amplitude, not the period (eq. 1 below). Hence, period behavior is independent of whether the parameter changes slowly or instantaneously.

Nonetheless, we agree that the reviewer raised an important point in that the situation may differ during oscillation onset, where no period modulation was visible. Therefore, we put further effort into simulating oscillation onset for slowly varying bifurcation parameter of either a SNIC or a supercritical Hopf bifurcation, as indeed suggested by the reviewer. We used the model:

$$\begin{aligned}\frac{dx}{dt} &= x(\beta - x^2 - y^2) - 2\pi y(1 - \lambda y), \\ \frac{dy}{dt} &= y(\beta - x^2 - y^2) + 2\pi x(1 - \lambda y), \quad \forall x, y \in \mathbb{R},\end{aligned}$$

where x and y are two variables describing the state of the oscillator, and β and λ are the Hopf and SNIC parameters, respectively. Default values for β and λ were 1 and 0, respectively. The model was integrated using the ODE solver in the Scipy package (v1.3.1, Jones et al., 2001) in python ("from scipy.integrate import odeint"). Stochastic simulations were performed by using the Euler-Maruyama method.

The model can be better understood when the system is transformed into polar coordinates, i.e.

$$\begin{aligned}\frac{dr}{dt} &= r(\beta - r^2), \\ \frac{d\theta}{dt} &= 2\pi(1 - \lambda r \sin \theta), \quad \forall r, \theta \in \mathbb{R}^+, \quad (1)\end{aligned}$$

where $r = \sqrt{x^2 + y^2}$ and $\theta = \tan^{-1}(y/x)$ are the amplitude and phase of the system. For fixed parameter values, this system shows oscillations for positive values of β and $|\lambda r| < 1$, with maximum amplitude given by the radius of the limit cycle, $r_{LC} = \sqrt{\beta}$. At the limit cycle, for a fixed β , the period of the oscillator is given by

$$T = \frac{1}{\sqrt{1 - \beta\lambda^2}}.$$

Hopf simulations

To simulate the effects of a Hopf bifurcation under a slowly changing parameter, Equation 1 was simulated with a fixed value of $\lambda = 0$ and

$$\beta(t) = k_\beta t,$$

where k_β is the rate of change of β . For illustrative purposes, the value of $\beta(t)$ was defined to be 1 when $\beta(t) > 1$. All deterministic simulations for a slowly changing β were performed with initial conditions of $r_0 = 10^{-5}$ and stochastic simulations with a value of $r_0 = 0$. The initial phase was defined to be $\theta_0 = \frac{\pi}{2}$.

Solutions for the amplitude go through an interval where the solution remains close to the steady state and then jumps suddenly to a neighbourhood of the limit cycle. However, the amplitude approaches asymptotically the limit cycle and thus, the system was determined to have reached the limit cycle if the difference between the rate of change of the radius of the limit cycle and the amplitude was sufficiently small, i.e.,

$$\left| \frac{dr}{dt} - k_\beta \right| < \phi.$$

For the simulations, the threshold $\phi = 0.01$.

SNIC simulations

To simulate the effects of a SNIC bifurcation under a slowly changing parameter, Equation 1 was simulated for a value of $\beta = 1$ and

$$\lambda(t) = 1 - k_\lambda t,$$

where k_λ is the rate of change of λ . As the effect for λ is symmetric, values were constrained to the positive real numbers including zero. Negative values were set to zero.

The system was initialized at the SNIC bifurcation point on the limit cycle, i.e., the initial conditions for phase and amplitude were defined to be $\theta_0 = \frac{\pi}{2}$ and $r_0 = 1$, respectively.

As we show in the new Fig. EV3, for a slowly varying SNIC parameter λ , we observe the expected long period that approaches a shorter constant period over time. By contrast, if the bifurcation parameter changes rapidly or instantaneously, oscillation onset will occur with a (nearly) stable period. In other words, if λ changes sufficiently rapidly, oscillation onset may not be accompanied by a noticeable change in period over time for a SNIC bifurcation.

For a slowly varying Hopf parameter β , oscillation may occur with a substantial delay relative to the bifurcation, as correctly predicted by the reviewer (new Fig EV3). The extent of this delay is strongly affected by the initial conditions, and specifically the value of r_0 (the smaller the value, the longer the delay.) Once oscillations occur, they may rapidly approach size r_{LC} (i.e., reach the envelope), which, depending on the rate and duration of parameter change, may continue to increase. Notably, irrespective of the rate of parameter change, i.e., even for an instantaneously changing parameter, oscillations always initiated with a transient, where $r \ll r_{LC}$. In principle, this could be used to distinguish experimentally between a SNIC and a supercritical Hopf bifurcation. However, in practice, this is difficult because it would require a somewhat arbitrary decision on when exactly the first oscillation is detected, and, in our specific case, because TC1 only starts at 5h.

For practical reasons, a distinction between a supercritical Hopf and a SNIC bifurcation during onset of the *C. elegans* oscillator based on amplitude and period behaviors is thus not

readily feasible. However, as we pointed out in the manuscript previously, the two bifurcations also differ in the stable state, which is on the limit cycle for a SNIC but distinct from the limit cycle for a supercritical Hopf bifurcation. Our data support that the stable state coincides with a particular phase of the oscillator, and thus an arrest on the limit cycle, which is a feature specific to a SNIC bifurcation.

To explore this further, and going beyond the reviewer's request, we asked whether it would be possible to achieve phase control with a supercritical Hopf bifurcation such that oscillations would always start from the same phase (although the stable state would differ). We found that this was not possible, at least without invoking additional mechanisms to avoid or compensate for noise. (And we note that an absence of noise appears not only biologically unrealistic, but it would also massively increase the delay between bifurcation and observed oscillation onset. However, we can constrain this time in our system to the time between plating and first observation of oscillation, and thus to $< 1T$.) Based on all these considerations, we think that a SNIC bifurcation for oscillation onset is the most parsimonious explanation of our data, and we thank the reviewer for prompting us to examine these alternatives to strengthen our case.

6. I suggest to write subcritical Hopf instead of supH.

We have followed the advice and substituted supH with supercritical(!) Hopf

7. Pearson correlation is sensitive to outliers. A brief comparison with Spearman correlation would be interesting.

We had also been concerned about this possibility but found in our initial analysis that the two approaches yielded highly similar results for our data. We have now added this analysis to Fig. S8.

8. page 39: Why 13 degrees of freedom? In my eyes, harmonic regression and trend elimination requires less parameters.

The reviewer is right that we only need one parameter for the sine and one parameter for the cosine. However, degrees of freedom are the number of data points that go into the estimation of the parameters used after taking into account these parameters, in our case yielding: $df = 16 \text{ time points} - 2 \text{ parameters} - 1 \text{ (sample mean)} = 13$.

9. page 40, line 11: Underlined T?

We fixed this, thank you.

Reviewer #3:

This is an engaging paper that builds on prior work by this group on the gene expression oscillations that occur during the various developmental stages of *C. elegans*.

The authors first extend their previous work on developmental oscillations in this system by

assaying at all stages of development. This shows that oscillations occur at every stage of development. In this study, they were able to then "glue" the gene expression profiles from the larval stages together to form a comprehensive dataset that could be used to track 3680 oscillating transcripts through the whole time course. Because they also assayed gene expression before oscillations began, and as they terminated.

The authors then sought to explain the observed gene expression cycle behaviors by examining two potential models: a superficial Hopf bifurcation and a SNIC. Clearly, the oscillations fitted with a SNIC bifurcation, given that their amplitude remained high and period lengthened in the final oscillation prior to adulthood (and transition to non-oscillatory behavior).

In general, the paper is welcomed development of the group's previous findings, but there are some issues that should be addressed prior to publication, particularly with respect to some exploration of mechanism.

We thank the reviewer for his/her favorable evaluation of our manuscript. We agree that this work is a starting point for further mechanistic exploration, although we expect that this requires work of a much larger scale than what is feasible during a revision

Major points

1. There is a lot of correlation in this paper, and although the SNIC model does move towards mechanism, the authors don't really talk about how / why oscillations begin/end. In particular, there is a latency between putting the worms on food and the oscillations starting. Is the starvation, followed by feeding, the cue that starts the oscillations (albeit with a phase delay)? Does gene ontology etc. reveal anything interesting in this regard? It would be much more informative if the authors try to address this by looking specifically into metabolic connections (which seem likely), e.g. in the context of Crosby et al. Cell 2019 (PMID: 31030999) and insulin signaling and its relationship to the circadian clockwork. Something similar might be involved here.

Our manuscript is the first one to describe that oscillations have a beginning and an ending, and clearly defined ones as well. Accordingly, there is currently no information on what could be causing either, although we may speculate. The impact of insulin on the circadian clock to which the reviewer refers is very intriguing, although we understand that this signaling modulates, rather than starts or stops circadian oscillations. Adding to this, based on current (admittedly limited) knowledge, there seem few, if any, direct parallels between circadian clocks and the *C. elegans* oscillator. This said, as we pointed out in the Discussion, *C. elegans* developmental checkpoints have been reported to involve IGF signaling, so we agree that this is a potentially highly relevant and interesting pathway.

Obviously, investigating such a speculative link in detail will require extensive experimentation, well beyond the scope of this work and a revision. Nonetheless, we have followed the reviewer's advice and explored whether specific pathways or functions are enriched during specific times of oscillations. As s/he suggested, we performed gene ontology (GO) analysis in a time resolved manner (i.e. on genes binned by their calculated peak phase), either for Biological Processes (Figure Panel A below) or Molecular Functions (Figure Panel B below).

Although this revealed enrichments of certain terms at specific times (shown in the figure below) as previously observed (Hendriks et al., Mol Cell 2014), it did not appear to provide any hint as to mechanisms of oscillator arrest or re-initiation. In particular, IGF-signaling related terms were not enriched at any time. However, we point out that GO annotations tend to lack sufficient granularity in *C. elegans* so that (in our experience) this approach often works rather poorly in the worm. Case in point, the GO-term annotations of DAF-2, the key IGF receptor, are: reproduction, nucleotide binding, immune system process, protein kinase activity, protein tyrosine kinase activity, transmembrane receptor protein tyrosine kinase activity but neither insulin nor IGF-signaling.

Given these limitations of GO term enrichment analysis, we performed further analyses specifically on 312 genes peaking around TP 19 and 211 genes with troughs around TP19, using two additional tools, WormPath and WormExp, and we also specifically analyzed the patterns of known IGF-signaling pathway genes.

Wormpath permits pathway analysis of differentially expressed genes, using information on genetic interactions that is available on Wormbase. Input for the algorithm is a list of differentially expressed genes with their p-value. Since our data are not from a differential gene expression analysis (but determination of whether or not they fall in the oscillating category), we used the same p-value for all genes. *daf-2* comes up, yet the sets of 312 and 211 genes give poor statistics in general, preventing us from drawing any conclusions.

WormExp categorizes genes based on their differential gene expression analyses seen with previous, specific manipulations, such as those related to dauer & insulin signaling and food (category: DAF/insulin/food). Gratifyingly, 93 of the peaking genes are categorized as “up by starved (Mueller paper)” and 122 genes of trough genes “down by starved (Mueller paper)”. Although these responses to starvation are what we would predict, we think that this analysis lacks robustness for our dataset because it is based on a single study (i.e., categories are assigned based on gene expression papers from previous, individual papers, in this case doi: 10.1038/ncb3071), and, more importantly, samples in that study were collected at only a single time point. In other words, if the timing between food and no-food conditions were off (which we would expect would be difficult to avoid), oscillating genes would show up as differentially expressed, but the effect could be technical or biological. Hence, we lack confidence in this result.

To look more specifically at Insulin signaling, we investigated IGF signaling pathway genes (as annotated in Wormbook, [10.1895/wormbook.1.164.1](https://www.wormbook.org/book/doi/10.1895/wormbook.1.164.1)). Out of 18 genes, we had categorized only one, *akt-1*, as oscillating. *daf-2* and *daf-16* show low-grade oscillations based on visual inspection, but were not annotated as oscillating. While we consider this observation on *daf-2* and *daf-16* intriguing, these results as such obviously do little to support a specific role of insulin signaling in modifying oscillations.

Finally, we decided to investigate insulin-like peptides (ILPs) and related neuropeptides. Among 31 *flp*, 41 *nlp*, and 40 *ilp* genes in the *C. elegans* genome, many of which were sufficiently expressed in our dataset (30 *flp*, 37 *nlp*, and 31 *ilp* genes), we had categorized few as oscillating (0 *flp*, 11 *nlp* and 5 *ilp* genes). We do note that this small set has interesting (i.e., biased) phase distributions as shown in the bee-swarm plot provided below. We conclude that this approach fails to provide statistical evidence for an involvement of insulin signaling in oscillation offset, but that the few oscillating genes observed might be worth following up.

We conclude that currently available data do not provide any substantial evidence for or against an involvement of IGF signaling in oscillation onset or offset. Beyond the technical issues mentioned above, this may be owed to two major conceptual limitations of these types of gene mining approaches: First, they assume that a substantial fraction of oscillating genes encodes oscillator components rather than output. We consider this unlikely and suspect that in fact output may drown any signal from the core clock and its regulator. Second, they assume that clock components or modifiers achieve rhythmic activity through oscillatory transcription. We would not expect this a priori for a signaling pathway, and also in the paper referenced by the reviewer, PER regulation by insulin occurs translationally. Hence, understanding what drives oscillation onset, offset and developmental checkpoints mechanistically will require dedicated experimental dissection, which is outside the scope of the present work.

2. The classification of genes as rhythmic is based on a cosine fitting algorithm / linear modelling. The authors do not seem to have used robust statistics here (using confidence interval only as far as I can see). There is no mention of multiple-testing correction and false discovery rate estimations for the classifications. The authors need to carry out and state what cut-offs they have used, perhaps using more widely used algorithms for temporally dense datasets (e.g. metaCycle, Cyclops, JTK_Cycle, RAIN). These alternative classifications will yield more robust statistics and FDR estimates. Also, these algorithms should allow the authors to assess period ranges (e.g. 4-9 hour period genes) rather than constraining the fitting to a 7 hour period.

The reviewer's comments are well taken and we apologize for our lack of clarity. We have now indicated cut-offs more clearly in the main text. In addition, and as proposed by the reviewer, we analyzed our data with the Meta2D function from the package MetaCycle in R. The results overlapped most extensively with those of our cosine wave fitting analysis (Review Figure 1A & B, below), both in terms of oscillating gene identities and amplitudes, and genes observed in only one of the two methods typically had a low amplitude (new Fig. S1). Based on visual inspection of the resulting heatmaps of uniquely identified genes (Review Figure 1C and D, below), we would suggest that the cosine wave fit provides the better result, but given the massive overlap of the results, this difference is negligible for our analyses.

Also as proposed by the reviewer, we used Meta2D to challenge our period determination. Using a period window of 4–9 hours, we observed the expected 7-hour period for the majority of genes identified by cosine wave fitting. A virtually identical distribution occurred for the genes identified by Meta2D (new Fig. S1). Moreover, we repeated the cosine wave fits with different periods (5, 6, 6.5, 7, 7.5, 8 and 9 hours) and found that the results were highly consistent for periods between 6 hours and 8 hours, i.e., similar numbers of mostly overlapping genes were identified. By contrast, numbers declined sharply for periods outside these boundaries (Review Figure 2, below). We conclude that the cosine wave fitting approach works robustly and enabled identification of an appropriate data set.

A

Cosfit oscillating genes (n = 3,739)

**B**

Meta2D oscillating genes (n = 3,628)

**C**

Cosfit unique oscillating genes (n = 200)

**D**

Meta2D unique oscillating genes (n = 89)

Review Figure 1: Comparison of oscillating genes identified by Cosine fitting and Meta2D analysis

(A,B) Mean-normalized gene expression heatmaps of oscillating genes identified by Cosine fitting (A) or Meta2D (B).

(C) Mean-normalized gene expression heatmaps of oscillating genes identified by Cosine fitting, but not by Meta2D (cosfit unique).

(D) Mean-normalized gene expression heatmaps of oscillating genes identified by Meta2D, but not by Cosine fitting (Meta2D unique)

Genes were sorted by peak phase determined in the corresponding method.

Review Figure 2: Fitting cosines with different periods

The oscillating genes identified by fitting a cosine with a period of 7 hours (purple) compared to the oscillating genes identified by fitting a cosine with a 5h, 6h, 6.5h, 7.5h, 8h and 9h period, using cut-offs as in Fig. 1 of the manuscript. The number of genes shared between both periods (black) and the number of genes unique to either period (different colors) are indicated.

3. The discussion is way too long. It should be cut to a third of the size.

We appreciate the reviewer's point and have revised and shortened the discussion. At the same time, we wish to point out that we are dealing with different audiences here that may care about different take-home messages. In particular, although our manuscript has a strong "quantitative" focus, we think it has many relevant insights for a *C. elegans* community that may be less familiar with the analyses that we have used. Hence, we have decided for a more targeted revision instead of the more radical approach advocated by the reviewer.

Minor points

1. The authors use Hilbert transforms on a few occasions. They should give a bit of background to non-specialists about why this was used and what it does to the data.

We have followed this suggestion and provided a brief description of Hilbert transforms in the Appendix.

5th Jun 2020

Manuscript Number: MSB-20-9498R

Title: Developmental function and state transitions of a gene expression oscillator in *C. elegans*

Author: Milou Meeuse

Yannick Hauser

Lucas Morales Moya

Gert-Jan Hendriks

Jan Eglinger

Guy Bogaarts

Charisios Tsiiris

Helge Großhans

Thank you for sending us your revised manuscript. We have now heard back from the two reviewers who were asked to evaluate your study. As you will see the reviewers are satisfied with the modifications made and think that the study is now suitable for publication.

Before we can formally accept your manuscript, we would like to ask you to address a few remaining editorial issues listed below.

REFEREE REPORTS

Reviewer #2:

I raised two points - differences between sliding parameter changes and true bifurcations and the illustration of claims by simulations of normal form models. Both aspects have been addressed very convincingly. Thus the already quite exciting paper has been improved properly.

Reviewer #3:

The authors have done a very good job in revising this manuscript and clarifying a number of issues that were raised in the initial review. It is therefore suitable for publication now that they have addressed the important issues with further analyses.

Corresponding Author Name: Helge Grosshans

Manuscript Number: MSB-20-9498RR